# Diagnosing Multi-step Reasoning Failures in Black-box LLMs via Stepwise Confidence Attribution

**Xiaoou Liu** [1]  **Tiejin Chen** [1]  **Dengjia Zhang** [2]  **Yaqing Wang** [3]  **Lu Cheng** [4]  **Hua Wei** [1]

## Abstract

Large Language Models have achieved strong performance on reasoning tasks with objective answers by generating step-by-step solutions, but diagnosing where a multi-step reasoning trace might fail remains difficult. Confidence estimation offers a diagnostic signal, yet existing methods are restricted to final answers or require internal model access. We introduce Stepwise Confidence Attribution (SCA), a framework for closed-source LLMs that assigns step-level confidence based only on generated reasoning traces. SCA applies the Information Bottleneck principle: steps aligning with consensus structures across correct solutions receive high confidence, while deviations are flagged as potentially erroneous. We propose two complementary methods: (1) NIBS, a non-parametric IB approach measuring consistency without graph structures, and (2) GIBS, a graph-based IB model that learns subgraphs through a differentiable mask to capture logical variability. Extensive experiments on mathematical reasoning and multi-hop question answering show that SCA reliably identifies low-confidence steps strongly correlated with reasoning errors. Moreover, using step-level confidence to guide self-correction improves the correction success rate by up to 13.5% over answer-level feedback.

## 1. Introduction

The ability to diagnose where a multi-step reasoning trace fails is essential for improving the reliability of large language models (LLMs). Solution traces such as Chain-of-Thought (CoT) (Wei et al., 2022) or Graph-of-Thought (GoT) (Besta et al., 2024) provide transparency into model reasoning, yet intermediate errors remain hard to identify and can critically affect the final prediction. Recent work has explored step-level diagnostics for reasoning traces, largely falling into two categories. The first trains supervised classifiers with step-by-step human annotations to label whether each reasoning step is correct (Jiao et al., 2025; Zheng et al., 2024; Lightman et al., 2023). The second prompts the LLM itself as a judge to critique each solution step by step (Weng et al., 2023; Li et al., 2024). While both directions can provide useful signals, the former requires expensive human annotation, and the latter inherits bias and inconsistency from the judge model (Chen et al., 2026a), limiting scalability and reliability.

Confidence estimation (CE) provides a complementary direction for assessing reliability, as it can operate directly on model outputs (Liu et al., 2025). Prior work has shown that measures such as semantic variance across sampled generations (Lin et al., 2023; Golovneva et al., 2022) or predictive entropy over logits (Lin et al., 2024; Kuhn et al., 2023) provide informative signals for estimating whether a final answer is correct. However, restricting CE to the final answer yields only a coarse reliability signal and fails to indicate which specific step in a reasoning trace is responsible for an error. This limitation motivates the need for Stepwise Confidence Attribution (SCA), where the goal is to assign confidence scores to individual reasoning steps and thereby provide fine-grained diagnostic signals.

Extending confidence attribution from the final-answer level to the step-wise setting introduces a key challenge. Reasoning traces generated by LLMs exhibit substantial *output variability*: correct solutions may differ in step order, expression, or level of detail. Yet, despite these surface-level differences, correct reasoning paths are not random; they are governed by the underlying logic of the problem. Consequently, valid traces tend to converge on specific *key logical invariants*, i.e, critical intermediate states or semantic milestones that are necessary to derive the correct answer (e.g., calculating intermediate costs in a math problem). A robust SCA must distinguish this legitimate variability from true errors. While legitimate variability represents alternative trajectories between these logical anchors, errors act as deviations that drift away from the consensus of valid reasoning.

---
[1]Arizona State University [2]Johns Hopkins University [3]Purdue University [4]University of Illinois Chicago. Correspondence to: Hua Wei <hua.wei@asu.edu>.

*Proceedings of the 43rd International Conference on Machine Learning*, Seoul, South Korea. PMLR 306, 2026. Copyright 2026 by the author(s).

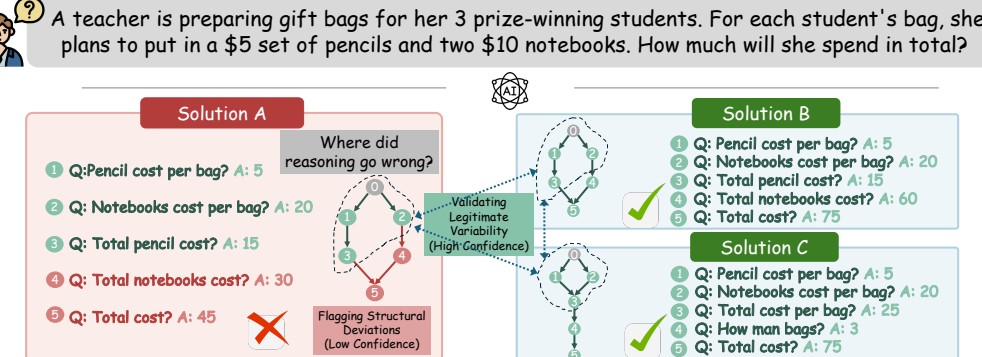

*Figure 1.* Example of reasoning trace variability in GSM8K dataset. Two distinct solution paths (B and C) yield the same correct answer, while another path (A) contains an erroneous step leading to a wrong result. Stepwise confidence attribution needs to distinguish legitimate variability from true logical inconsistencies.

For example, in Figure 1, Solutions B and C arrive at the correct answer through different computation orders. Despite the permutation of steps, both solutions explicitly resolve the necessary intermediate values (the cost of pencils and notebooks). They share functional equivalence in their reasoning logic and should receive high confidence. In contrast, solution A, while structurally similar to B, introduces an incorrect operation at step 4; this deviation undermines the common structure and should receive low confidence.

Our key idea is to aggregate multiple solution traces across correct solutions and identify logical invariants. These logical invariants serve as anchors of reliable reasoning and are assigned high confidence, while steps absent from consensus patterns are assigned low confidence, since they are more likely to reflect spurious or error-prone reasoning. Consensus patterns thus act as a proxy for the latent logical pattern underlying a problem, enabling fine-grained and robust confidence attribution.

This intuition of finding a shared pattern amidst noisy variations maps naturally onto the **Information Bottleneck (IB)** principle. IB provides a formal language for balancing two competing objectives: compressing the input trajectory by discarding non-essential variations (the compression term), while retaining maximal information about the underlying correct reasoning pattern (the relevance term). Here, the input $X$ is a reasoning trajectory composed of multiple steps, the compressed representation $Z$ is the trajectory expressed with confidence weights on its steps, and the target $Y$ denotes the correctness signal of the trajectory. The objective is $\min_Z I(X; Z) - \beta I(Z; Y)$, where the first term encourages compression by selecting only a sparse subset of steps, and the second term ensures that this subset retains critical information required for correctness. Within this framework, we explore two complementary instantiations:

- **Non-parametric IB for Stepwise Confidence.** $Z$ is realized as a set of consensus steps derived from correct solutions, and step-level confidence is obtained by measuring how well a trajectory aligns with this set.

- **Graph IB for Stepwise Confidence (GIBS).** To better handle structural variability, trajectories are represented as graphs, and $Z$ is a subgraph selected through a differentiable mask. Confidence scores are produced by aligning the selected subgraph with correctness signals, providing a more flexible treatment.

Empirical results show that both IB-based methods produce accurate confidence estimates. Beyond diagnostics, we demonstrate the impact of fine-grained CE on downstream LLM performance, showing that using step-level signals guides self-correction and improves correction success rate by up to 13.5% over answer-level feedback. Ablation studies verify the contribution of each component, and robustness analyses demonstrate label availability conditions, generalization across reasoning formats, and domain shift.

## 2. Related Work

**Confidence Estimation in LLMs.** Confidence estimation (CE) aims at assigning reliability scores to individual outputs (Liu et al., 2025). Most CE methods focus on the final answer, either through internal signals such as entropy (Kuhn et al., 2023; Lin et al., 2024; Patel et al., 2026), or through black-box signals such as agreement across sampled outputs (Lin et al., 2023; Chen et al., 2025; Da et al., 2025; Chen et al., 2026b). However, these methods give no insight into intermediate steps. Recent stepwise CE approaches (Ye et al., 2025; Han et al., 2025) attempt to assign confidence along reasoning traces, but require internal access to token probabilities, restricting them to open-source models. Some works also model reasoning as graphs (Besta et al., 2024; Pandey et al., 2025), but the representation itself is orthogonal to our problem setting. Our work differs by introducing the problem of stepwise confidence attribution in the black-box setting, which provides scalable confidence signals at the step level using only generated reasoning traces.

**Reasoning Verification in LLMs.** Reasoning verification studies whether multi-step reasoning is correct. Reasoning verification is categorized into answer-level and step-level approaches. Answer-level methods, such as self-consistency (Wang et al., 2022), outcome-based verifiers (Uesato et al., 2022; Zhang et al., 2024), and LLM judges (Li et al., 2024), often lack process diagnosability (Tyen et al., 2023). Step-level methods assess intermediate steps via human supervision (Lightman et al., 2023; Zheng et al., 2024), automated rewards (Wang et al., 2023a; Setlur et al., 2024), or graph structures (Cao, 2023; Fang et al., 2025; Mukherjee et al., 2025). However, these rely on costly annotations (Lightman et al., 2023) or unreliable subjective judgments (Szymanski et al., 2024; Stechly et al., 2024; Jacovi et al., 2024). We address this by introducing quantitative confidence signals to reduce cost and bias.

## 3. Problem Statement

Standard Answer-level Confidence Estimation (CE) assigns a reliability score to the final output $A$. (For a formal definition of Answer-level CE and distinctions between white-box and black-box settings, please refer to Appendix B.) While effective, answer-level CE provides only a coarse signal, failing to reveal *which reasoning steps* contribute to success or failure. This limitation is critical in high-stakes decision-making where identifying the location of error is essential.

To address this, we shift granularity to the intermediate reasoning process. For reasoning tasks, models generate a *reasoning trajectory* $y_i = (T_i, A_i)$, where $T_i = \{t_{i1}, \ldots, t_{iL_i}\}$ is a sequence of steps. We introduce *Stepwise Confidence Attribution (SCA)*, utilizing final-answer correctness to anchor step-level reliability. It should be noted that in this paper, we distinguish between model certainty and step reliability. In a white-box setting, a step may be generated with high token probability (high certainty) yet be logically flawed; conversely, a correct step may have low probability if it is novel. Therefore, in this work, we refer to "confidence" not as subjective generation probability, but as a *reliability score* quantifying a step's contribution to a correct answer.

**Problem 1** (Stepwise Confidence Attribution). *Given an input $x$ and $N$ sampled trajectories $\mathcal{S} = \{(T_i, A_i, z_i)\}_{i=1}^N$ with correctness labels $z_i \in \{0, 1\}$, the goal is to learn a mapping $f : (T_i, \mathcal{S}) \to \{c_{ij}\}_{j=1}^{L_i}$, where $c_{ij}$ is the confidence score assigned to step $t_{ij}$. Here, $c_{ij}$ reflects the step's alignment with common structures across correct trajectories, serving as a proxy for correctness attribution.*

The primary goal of SCA is diagnostic. For an incorrect trajectory ($z_i = 0$), $f$ should assign low confidence to steps responsible for the error. For a correct trajectory ($z_i = 1$), steps matching the logical consensus of valid solutions should receive high confidence. While this assumes final-answer labels, such signals are naturally available in evaluation scenarios (Liu et al., 2023; Augenstein et al., 2024; Gao et al., 2025; Zhao et al., 2024; Wang et al., 2023b), avoiding the need for costly step-level annotations. We also show possible adaptations for label-free settings in Section 5.5.

## 4. Method

Extending confidence estimation to the stepwise setting introduces the core challenge of distinguishing benign output variability from true reasoning errors. To tackle this challenge, our approach is rooted in a key insight: while individual correct solutions may vary on the surface, correct traces often share a latent common structure that reflects the essential reasoning pattern. A robust attribution method must therefore learn to identify this shared structure and assign confidence scores based on each step's consistency with it. Our method captures these latent structures by constructing consensus anchors from correct reasoning traces. We leverage these consensus anchors to guide confidence attribution, formalizing the problem under the **Information Bottleneck (IB)** principle.

### 4.1. Information Bottleneck Formulation

We cast stepwise confidence attribution as an instance of the Information Bottleneck (IB) principle. Given a trajectory $T_i = \{t_{i1}, \ldots, t_{iL_i}\}$ sampled from the LLM, the goal is to produce a confidence mask $Z = \{c_{ij}\}_{j=1}^{L_i}$ over its steps. Unlike settings with step-level annotations, we do not observe the correctness of individual steps. Instead, we only have access to final-answer labels $z_i \in \{0, 1\}$ for each trajectory. The target $Y$ must therefore be derived from the sampled set $\mathcal{S} = \{(T_i, A_i, z_i)\}$. Concretely, $z_i$ partitions $\mathcal{S}$ into correct and incorrect subsets, and consensus anchors $\mathbf{m}_{ij}$ are aggregated from $\mathcal{S}_{\text{correct}}$ to approximate the latent reasoning structure. The IB objective is

$$\min_Z \ I(T_i; Z) - \beta I(Z; Y), \qquad (1)$$

where $I(T_i; Z)$ encourages *compression* by sparsely selecting steps, and $I(Z; Y)$ ensures *relevance* between the retained steps and correctness signals inferred from consensus. In the following, we instantiate this principle with two variants: (1) **NIBS**, a non-parametric IB approach, and (2) **GIBS**, a graph-based model that applies a differentiable IB mask to handle structural variability.

### 4.2. Non-parametric IB for Stepwise Confidence (NIBS)

Directly solving the IB objective in Eq. 1 is generally intractable, since it requires searching over all possible confidence values of steps in $T_i$ and estimating mutual information terms. A natural approximation is to assume that steps consistently appearing in correct trajectories are the most

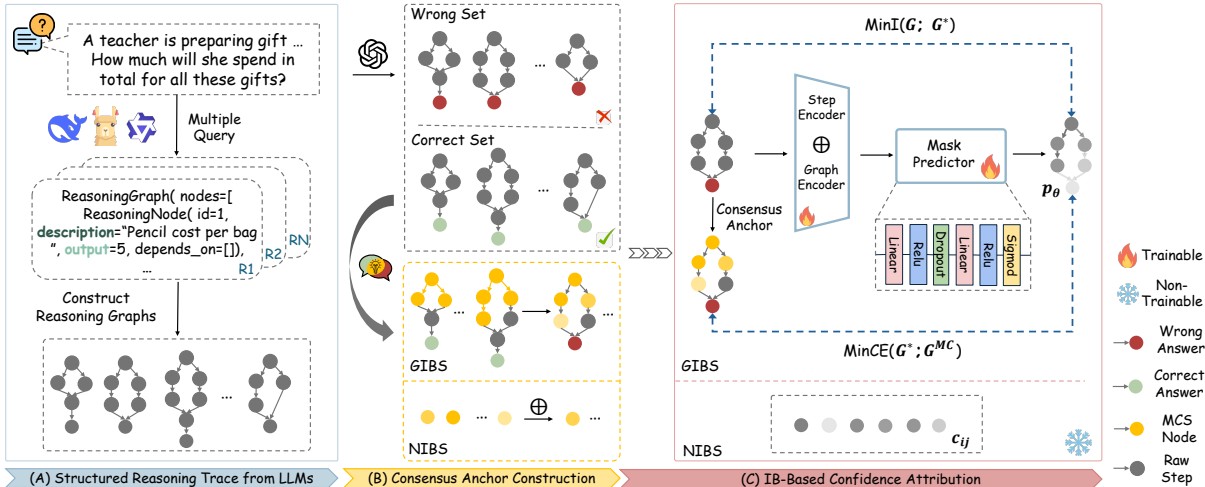

*Figure 2.* Overview of the IB-based stepwise confidence attribution framework. The process consists of (A) constructing structured reasoning traces from LLM outputs, (B) deriving consensus anchors from correct trajectories, and (C) applying the IB formulation through NIBS and GIBS to produce confidence scores.

informative about correctness $Y$, while steps absent from correct trajectories carry little predictive value. Under this approximation, the IB objective reduces to retaining consensus steps as the compressed representation $Z$, and assigning higher confidence to steps that align with this consensus. Formally, given a trajectory $T_i = \{t_{i1}, \dots, t_{iL_i}\}$ and one of its steps $t_{ij}$, its confidence can be computed as

$$c_{ij} = \mathbb{E}_{S \sim \mathcal{S}_{\text{correct}}} \Big[ \text{Agg}\Big( \{\text{sim}(\mathbf{t}_{ij}, \mathbf{t}') \mid \mathbf{t}' \in S\} \Big) \Big], \quad (2)$$

where $\text{sim}(\cdot, \cdot)$ measures semantic similarity between steps (e.g., cosine (Golovneva et al., 2022) or NLI (He et al., 2020)), and Agg aggregates similarities within a trajectory (e.g., maximum or mean).

This construction instantiates the IB principle: compression $I(T_i; Z)$ is realized by restricting $Z$ to consensus steps, while relevance $I(Z; Y)$ is maximized because consensus overlap is strongly correlated with correctness. NIBS provides a closed-form, non-parametric, and training-free solution to the IB objective. However, NIBS ignores structural dependencies by matching steps based on the semantic similarity alone, potentially grouping nodes that are structurally disparate. This limitation motivates a graph-based IB formulation with learned subgraph selection.

### 4.3. Graph IB for Stepwise Confidence (GIBS)

While NIBS provides a simple closed-form instantiation of the IB principle, it ignores structural dependencies among reasoning steps. To capture such dependencies, we adopt a graph-based formulation in which each trajectory $T_i$ is represented as a directed graph $G_i = (V_i, E_i)$. Nodes $V_i$ denote intermediate results, while edges $E_i$ encode logical dependencies or sub-questions that drive the reasoning forward. Each reasoning step $t_{ij}$ is thus represented as a pair

$(v_{ij}, e_{ij})$, where $e_{ij}$ is the sub-question or logical operation and $v_{ij}$ is the resulting intermediate answer. As illustrated in Figure 1, darker sentences (Q) represent edges and lighter sentences (A) represent intermediate results, together forming a reasoning step.

This representation allows step selection to account not only for surface similarity but also for the topology of reasoning. In this setting, $Z$ is instantiated as a selected subgraph $G^* \subseteq G_i$. Then the objective of Eq. 1 becomes

$$\min_{G^*} \ I(G_i; G^*) - \beta I(G^*; Y). \quad (3)$$

Since step-level labels are not observable, we approximate $Y$ by a consensus graph $G^{MC}$ aggregated from correct trajectories in the sampled set $\mathcal{S}$. For each reasoning graph $G_i$, we compute the Maximum Common Subgraph (MCS) (Mc-Creesh et al., 2017) between $G_i$ and each correct graph $G_k \in \mathcal{G}_{\text{correct}}$. Then, each MCS highlights the reasoning components shared between $G_i$ and a correct solution. Aggregating these pairwise MCS results yields $G^{MC}$, which serves as a consensus structure reflecting how $G_i$ aligns with correct reasoning. Details of MCS calculation and aggregation can be found in Appendix C. Replacing $I(G^*; Y)$ with $I(G^*; G^{MC})$, the structure-alignment IB objective now becomes:

$$\min_{G^*} \ I(G_i; G^*) - \beta I(G^*; G^{MC}). \quad (4)$$

**From IB to Soft-mask Relaxation.** Directly solving the IB objective on graphs is intractable: selecting a discrete subgraph $G^* \subseteq G_i$ is a combinatorial problem, and estimating mutual information terms between discrete subgraphs and correctness signals is not feasible in practice. We therefore introduce a differentiable mask $\mathbf{p}_\theta = \{p_{\theta, ij}\}$ over steps, yielding a soft subgraph $G^* = G_i \odot \mathbf{p}$ as the compressed

representation. For each step $t_{ij}$, the model predicts a selection probability $p_{ij} \in [0, 1]$ that reflects its contribution to the retained reasoning structure.

**Approximating the IB Objective.** The two mutual information terms in the IB objective are approximated with tractable surrogates:

*Compression.* We start from the mutual information identity $I(G_i; G^*) = H(G^*) - H(G^* \mid G_i)$. To rigorously bound this objective, we adopt the Variational Information Bottleneck framework. Given that our step-selection mask consists of binary decisions (keep or drop), we naturally model the variational prior $r(z)$ as an independent Bernoulli distribution parameterized by a small constant $\epsilon < 0.5$. Consequently, the compression objective is minimized via the KL divergence between the predicted mask $\mathbf{p}_\theta$ and this sparse prior:

$$
\begin{aligned}
\mathcal{L}_{\text{compress}} &\approx D_{KL}(\mathbf{p}_\theta \| r) \\
&= \sum_j \left[ p_{\theta,ij} \log \frac{p_{\theta,ij}}{\epsilon} + (1 - p_{\theta,ij}) \log \frac{1 - p_{\theta,ij}}{1 - \epsilon} \right].
\end{aligned} \quad (5)
$$

Mathematically, this term decomposes into a sparsity penalty and a negative entropy component. While some formulations of IB maximize entropy to learn a compressed stochastic representation, our goal is to select a single, determinate subgraph. By minimizing the entropy of the mask distribution (a component of the KL term), we force the model to make confident, binary-like decisions for each step (i.e., $p_{\theta,ij} \to 0$ or 1). This directly encourages a compressed representation of the reasoning graph, thus satisfying the compression objective $I(G_i; G^*)$ in a variational manner.

*Relevance.* To maximize the relevance term $I(G^*; Y)$, standard IB approaches typically require training an auxiliary classifier to estimate the conditional likelihood $p(Y|G^*)$, which is computationally expensive and unstable in black-box settings. Instead, we adopt a consensus-based surrogate strategy. We posit that the consensus structure $G^{MC}$, derived from the intersection of correct solutions, serves as a robust proxy for the essential reasoning information required to predict correctness ($Y = 1$). Consequently, we approximate the maximization of mutual information by directly minimizing the Cross-Entropy (CE) between the predicted mask $p_\theta$ and the consensus mask $m_i$: $\mathcal{L}_{rel} = CE(p_\theta, m_i)$.

**Training Objective.** For each reasoning graph $G_i$, the model $f_\theta$ outputs a soft mask $\mathbf{p}_\theta = f_\theta(G_i)$, where $p_{\theta,ij} \in [0, 1]$ denotes the selection probability for step $t_{ij} = (v_{ij}, e_{ij})$. The consensus mask $\mathbf{m}_i$ is obtained by aligning $G_i$ with the maximum common subgraph $G^{MC}$ constructed from correct trajectories. Each $m_{ij} \in \{0, 1\}$ indicates whether step $t_{ij}$ is part of the consensus reasoning structure. The final loss is

$$
\mathcal{L}(G_i) = H(\mathbf{p}_\theta) + \lambda \, \text{CE}(\mathbf{p}_\theta, \mathbf{m}_i). \quad (6)
$$

**Inference.** At test time, given a new reasoning graph $G$ without any gold final-answer labels, the model outputs a probability mask $\mathbf{p}_\theta = f_\theta(G)$. For each step $t_{ij} = (v_{ij}, e_{ij})$, the step-wise confidence score is $c_{ij} = p_{\theta,ij}$. Steps with high probabilities are considered reliable, while those with low probabilities are flagged as logically inconsistent or error-prone.

### 4.4. Implementation

Figure 2 illustrates the overall pipeline when we implement our SCA under the IB formulation.

**(A) Structured Reasoning Trace from LLMs.** To model the reasoning process, we move beyond linear sequences to explicit reasoning graphs. Specifically, we parse each trace into a reasoning graph $G_i = (V_i, E_i)$, where each step is represented as a node–edge pair $(v_{ij}, e_{ij})$, where $v_{ij}$ is the result and $e_{ij}$ is the operation producing it (Da et al., 2025; Chen et al., 2023; Amini et al., 2019; Da et al., 2024). For our experiments, we utilize LangFun-style prompting[1] to elicit these dependencies and extract graphs via a rule-based parser, though our framework is compatible with other parsing strategies (e.g., dependency parsing or implicit latent graphs). The full prompt is provided in Appendix D.5. A visualization is shown in Figure 1.

**(B) Consensus Anchor Construction.** To obtain the compressed representation $Z$, we identify steps shared across correct solutions. For NIBS, we operationalize this alignment by comparing every step to steps in the correct set via a semantic similarity function. In GIBS, consensus anchors are identified by solving a maximum common subgraph (MCS) problem (McCreesh et al., 2017) between candidate and correct graphs. Steps that consistently appear in the MCS are treated as high-confidence. See Appendix C for algorithms and Appendix C.4 for complexity analysis.

**(C) IB-Based Confidence Attribution.** NIBS computes confidence from consensus anchors in closed form without training. GIBS learns a parameterized mapping using a BERT encoder for semantic step features and a 2-layer GCN for structural context; these representations are fused to predict soft selection probabilities $p_\theta$ via the IB objective (Eq. 6). Note that while we adopt GCN based on preliminary empirical validation (see Appendix E for comparisons with other GNN architectures), these encoders remain replaceable components adaptable to different task complexities.

## 5. Experiments

We conduct comprehensive experiments to evaluate the effectiveness of our SCA methods.[2] Specifically, we address

---

[1] https://github.com/google/langfun
[2] Code can be found in https://github.com/Xiao0o0o/stepwise-confidence-attribution

| LLM | Llama3.1-8b | | | | DeepSeek-R1-Distill-Qwen-32B | | | | Phi4-reasoning | | | |
|---|---|---|---|---|---|---|---|---|---|---|---|---|
| Metrics | AUROC↑ | AUCPR↑ | ACC@80%↑ | ECE↓ | AUROC↑ | AUCPR↑ | ACC@80%↑ | ECE↓ | AUROC↑ | AUCPR↑ | ACC@80%↑ | ECE↓ |
| Dataset: GSM8K | | | | | | | | | | | | |
| P(true) | 0.4016 | 0.4711 | 0.5283 | 0.5504 | 0.5159 | 0.5820 | 0.5840 | 0.5802 | 0.5251 | 0.6956 | 0.6934 | 0.6851 |
| SL(norm) | 0.4790 | 0.5240 | 0.5513 | 0.2282 | 0.3700 | 0.5308 | 0.5262 | **0.1230** | 0.3851 | 0.5947 | 0.6799 | 0.2394 |
| Entropy | 0.4105 | 0.4962 | 0.5141 | 0.2518 | 0.5203 | 0.5550 | 0.5235 | 0.2583 | 0.4623 | 0.6795 | 0.6664 | 0.1648 |
| LECO | 0.3862 | 0.4586 | 0.5319 | 0.3783 | 0.3202 | 0.4110 | 0.4883 | 0.3395 | 0.2885 | 0.5735 | 0.6344 | 0.4021 |
| Cos-Max | 0.4537 | 0.5152 | 0.5513 | 0.3780 | 0.5269 | 0.5197 | 0.5676 | 0.3799 | 0.3494 | 0.5861 | 0.7010 | 0.1997 |
| Cos-Mean | 0.6078 | 0.6211 | 0.6175 | 0.3300 | 0.6633 | 0.6933 | 0.5748 | 0.3323 | 0.5959 | 0.7703 | 0.7199 | **0.1556** |
| NLI-Max | **0.7096** | **0.7890** | 0.5908 | **0.1162** | **0.7450** | **0.6982** | 0.6409 | 0.1456 | 0.6600 | 0.8141 | 0.7446 | 0.2009 |
| NLI-Mean | 0.5524 | 0.6103 | 0.5665 | 0.4376 | 0.6762 | 0.6508 | 0.6318 | 0.4189 | 0.5738 | 0.7704 | 0.7186 | 0.5527 |
| GIBS | 0.6910 | 0.7004 | **0.6292** | 0.2293 | 0.7289 | 0.6712 | **0.6532** | 0.2867 | **0.7892** | **0.8172** | **0.8117** | 0.3354 |
| Dataset: MoreHopQA | | | | | | | | | | | | |
| P(true) | 0.5228 | 0.5486 | 0.5450 | 0.5357 | 0.5177 | 0.6492 | 0.6606 | 0.6443 | 0.5086 | 0.6159 | 0.6362 | 0.6203 |
| SL(norm) | 0.4005 | 0.4506 | 0.5187 | 0.3168 | 0.2463 | 0.4894 | 0.6049 | 0.2829 | 0.3198 | 0.4966 | 0.5891 | 0.2751 |
| Entropy | 0.5510 | 0.5413 | 0.5586 | **0.2148** | 0.6103 | 0.7150 | 0.6812 | **0.0952** | 0.6012 | 0.6702 | 0.5709 | **0.0520** |
| LECO | 0.3280 | 0.4365 | 0.4921 | 0.3501 | 0.3116 | 0.5777 | 0.5642 | 0.3700 | 0.2760 | 0.4944 | 0.5782 | 0.4254 |
| Cos-Max | 0.3836 | 0.4446 | 0.5179 | 0.3687 | 0.3390 | 0.5129 | 0.6031 | 0.3066 | 0.3454 | 0.5284 | 0.6413 | 0.2614 |
| Cos-Mean | 0.5044 | 0.5487 | 0.5275 | 0.3160 | 0.5938 | 0.6919 | 0.6419 | 0.2434 | 0.4779 | 0.6492 | 0.6388 | 0.2023 |
| NLI-Max | 0.4937 | 0.5485 | 0.5347 | 0.2159 | 0.6663 | 0.7766 | 0.6457 | 0.1107 | 0.5801 | 0.6637 | 0.6961 | 0.2323 |
| NLI-Mean | 0.5124 | 0.5535 | 0.5314 | 0.3902 | 0.6291 | 0.6899 | 0.6498 | 0.4305 | 0.5440 | 0.6428 | 0.6776 | 0.4610 |
| GIBS | **0.6471** | **0.6694** | **0.5602** | 0.3173 | **0.8084** | **0.8357** | **0.7051** | 0.1832 | **0.6619** | **0.6866** | **0.7053** | 0.3560 |
| Dataset: Math | | | | | | | | | | | | |
| P(true) | 0.4584 | 0.4166 | 0.4551 | 0.4564 | 0.5055 | 0.6298 | 0.6226 | 0.6215 | 0.5277 | 0.7435 | 0.7554 | 0.7416 |
| SL(norm) | 0.5138 | 0.4627 | 0.4626 | 0.3250 | 0.4093 | 0.5433 | 0.6089 | 0.2319 | 0.3841 | 0.6597 | 0.7340 | 0.2220 |
| Entropy | 0.5120 | 0.4617 | 0.4550 | 0.2876 | 0.5190 | 0.6259 | 0.6225 | 0.1814 | 0.5190 | 0.7351 | 0.7479 | 0.2023 |
| LECO | 0.4378 | 0.4252 | 0.4382 | 0.2229 | 0.3838 | 0.5836 | 0.5901 | **0.1813** | 0.4089 | 0.7279 | 0.7044 | 0.3957 |
| Cos-Max | 0.4313 | 0.4021 | 0.4513 | 0.4408 | 0.3376 | 0.5015 | 0.5937 | 0.3093 | 0.3845 | 0.6617 | 0.7656 | 0.1568 |
| Cos-Mean | 0.5118 | 0.4848 | 0.4607 | 0.3946 | 0.5363 | 0.6583 | 0.6177 | 0.2463 | 0.6116 | **0.8503** | 0.7708 | **0.1013** |
| NLI-Max | 0.5173 | **0.5061** | **0.4743** | **0.1852** | 0.5310 | 0.6676 | 0.6340 | 0.2340 | 0.6043 | 0.8052 | 0.7925 | 0.3090 |
| NLI-Mean | 0.5148 | 0.4881 | 0.4670 | 0.3356 | 0.5407 | 0.6694 | 0.6267 | 0.4950 | 0.5739 | 0.8020 | 0.7712 | 0.5935 |
| GIBS | **0.5855** | 0.4890 | 0.4513 | 0.2737 | **0.5806** | **0.6831** | **0.6359** | 0.3786 | **0.6946** | 0.8078 | **0.8322** | 0.4050 |

*Table 1.* Overall results for step-wise confidence attribution on the GSM8K, MoreHopQA, and Math datasets. Our proposed method, GIBS, consistently outperforms baseline methods, especially on more complex reasoning tasks. The **best** and second-best results are highlighted. Higher AUROC, AUCPR, and ACC@80% and lower ECE indicate better performance.

the following research questions:

• **RQ1 (Accuracy)**: How accurately can our proposed methods (NIBS and GIBS) identify erroneous steps in a reasoning trace compared to strong baselines?

• **RQ2 (Utility)**: Can the stepwise confidence scores from our methods be used to improve the final-answer accuracy of LLMs through targeted self-correction?

• **RQ3 (Ablation & Robustness)**: What is the contribution of each component in the GIBS framework? How robust is our framework to variations in reasoning format, settings without final-answer labels, and domain shift?

## 5.1. Experiment Settings

Below we describe the experimental setup. Additional implementation details and sensitivity study are provided in Appendix D and Appendix E.

**Datasets.** We evaluate on three benchmarks for verifiable reasoning tasks, where final-answer correctness can be objectively determined: (1) GSM8K (Cobbe et al., 2021), a widely used math word problem dataset; (2) Math (Hendrycks et al., 2021), competition-level problems requiring more complex reasoning; and (3) More-HopQA (Schnitzler et al., 2024), a multi-hop QA dataset testing generalization beyond math.

**Compared Methods.** We compare the following methods: (i) four *white-box* CE approaches adapted to stepwise reasoning: SL(norm) (Lin et al., 2024; Cole et al., 2023), Token Entropy (Kuhn et al., 2023), P(true) (Kadavath et al., 2022), and LeCo (Yao et al., 2024); (ii) our proposed NIBS family, which instantiates the closed-form IB solution with different similarity functions and aggregation strategies; and (iii) our GIBS model, which learns to align subgraphs with consensus structures. To our knowledge, there are no existing black-box methods that directly address stepwise confidence attribution. We adapt two methods from related tasks (reasoning evaluation (Golovneva et al., 2022) and error identification (Mukherjee et al., 2025)) for comparison. Results in Appendix G show that our approach substantially outperforms these baselines.

**Configurations.** We evaluate three representative LLMs: LLaMA-3.1-8B-Instruct (Grattafiori et al., 2024), Phi-4-Reasoning (Abdin et al., 2025), and DeepSeek-R1-Distill-Qwen-32B (Guo et al., 2025). For each input, we sample $N = 20$ traces with temperature 1.0 to balance accuracy and diversity. For semantic similarity and equivalence, our framework is flexible in selecting a similarity function. We experiment with two approaches commonly used in UQ for LLMs: (1) cosine similarity between sentence embeddings from BERT (Devlin et al., 2019), and (2) semantic entailment predictions from an off-the-shelf NLI model (He et al.,

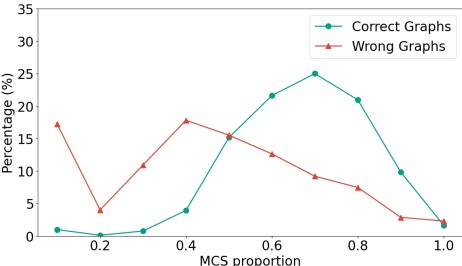

*Figure 3.* Distribution of average MCS proportion over 1,000 reasoning graphs. Correct graphs (green) concentrate near 0.8, while incorrect ones (red) peak near 0.4. This indicates that correct solutions are likely to follow a stable reasoning path.

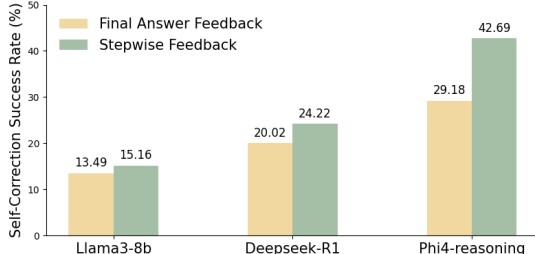

*Figure 4.* Effect of step-level feedback on correcting initially wrong answers in MoreHopQA. The baseline (yellow) provides only answer-level feedback, while our method (green) also highlights low-confidence steps.

2020; Lin et al., 2023). GIBS is trained on 2,000 reasoning graphs constructed from sampled solutions. At inference time, results for all methods are averaged over 10,000 trajectories per dataset. For final-answer correctness, mathematical datasets are evaluated by exact match with gold answers, while QA datasets use GPT-4o as a judge. Additional details on model selection, hyperparameters, prompts, and evaluation setup are provided in Appendix D.

**Metrics.** Following prior work in CE (Davis & Goadrich, 2006; Lin et al., 2023; Geifman & El-Yaniv, 2017), we adopt four complementary metrics: (i) **AUROC** and (ii) **AUCPR** evaluate ranking quality of confidence scores, with AUCPR being especially important under class imbalance since erroneous steps are sparse. (iii) **ACC@80%** measures selective prediction performance: we reject the 20% of predictions with the lowest confidence and report accuracy on the remaining 80%. (iv) **ECE** (Expected Calibration Error) assesses calibration by comparing predicted confidence with empirical accuracy across bins.

### 5.2. Accuracy of Stepwise Confidence Attribution

We first provide empirical support for the intuition underlying our framework: correct reasoning trajectories should exhibit stronger overlap in their common substructures. Figure 3 shows the distribution of average maximum common subgraph (MCS) scores across 1,000 reasoning graphs. Correct solutions concentrate around larger relative MCS sizes, while incorrect solutions peak around smaller values. This suggests that correctness is associated with stability and reproducibility of the reasoning path, while erroneous traces tend to diverge, producing fragmented structures. These observations provide empirical support for using consensus as a proxy signal for SCA.

We then compare NIBS and GIBS against strong white-box baselines for stepwise confidence attribution. Results in Table 1 show consistent patterns across models and datasets.

• *GIBS achieves the best AUROC in most settings.* Across three datasets and three LLMs, GIBS obtains the highest AUROC in 7 out of 9 configurations, demonstrating its strong ability to discriminate between correct and erroneous

steps. This advantage stems from explicitly modeling structural dependencies: by aligning candidate subgraphs with consensus anchors, GIBS captures reasoning patterns that local similarity methods miss. The improvement is particularly pronounced on MoreHopQA, where reasoning spans multiple passages and logical alignment becomes essential.

• *NIBS provides competitive performance without training.* Although NIBS variants only capture semantic overlap without explicit structural modeling, they still achieve strong results across all datasets. Notably, the performance of these variants is robust to the choice of external models used for computing semantic similarity, with Cos-Mean and NLI-Max achieving similar results. These results indicate that even non-parametric consensus alignment provides a reliable signal for stepwise confidence estimation.

• *Our framework generalizes to human-annotated step-wise benchmarks.* We also evaluate on PRM800K (Lightman et al., 2023), a well-established stepwise reasoning benchmark containing pre-collected GPT-4 solutions with free-form chain-of-thought reasoning and high-quality human step-level annotations (+1, -1, 0 for each step). Unlike our main experiments, PRM800K requires no additional parsing; each reasoning trace is directly represented as a linear graph where sentences are sequentially connected. As detailed in Appendix F.1, both NIBS and GIBS achieve strong performance on identifying incorrect steps, demonstrating the effectiveness of our approach on pre-collected traces with human-annotated ground truth.

### 5.3. Self-Correction with Stepwise Confidence

Previous experiments show we can detect errors; in this section, we show that this confidence attribution can actually guide effective self-correction. Specifically, we study whether highlighting low-confidence steps enables the model to successfully revise its erroneous answers. Our evaluation considers MoreHopQA instances that the LLM initially answered incorrectly, and asks the LLM to regenerate its answer once, using two forms of feedback: (1) **Final-answer feedback**: the model is only told that its final answer was wrong. (2) **Stepwise feedback**: in addition to

| Method | GSM8K | MorehopQA | Math |
|---|---|---|---|
| GIBS | **0.7892** | **0.6619** | **0.6946** |
| w/o Graph Encoder | 0.7228 | 0.6481 | 0.5961 |
| w/o Edge Encoder | 0.5186 | 0.3760 | 0.4763 |

*Table 2.* Performance of GIBS w.r.t. AUROC with and without the edge encoder or the graph encoder.

| | Llama3.1-8b | Deepseek | Phi4 |
|---|---|---|---|
| All trajectories | 0.5192 | 0.6479 | 0.5546 |
| Self-consistency | 0.5734 | 0.7843 | 0.6610 |
| Correct-only | **0.6471** | **0.8084** | **0.6619** |

*Table 3.* Comparison of consensus anchor strategies. Correct-only performs best, while Self-consistency provides effective weak supervision with comparable results.

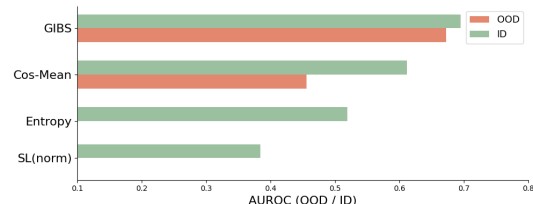

*Figure 5.* GIBS trained on MoreHopQA and tested on Math without re-training. GIBS consistently outperforms NIBS and white-box baselines under domain shift.

answer-level feedback, the model is shown its previous reasoning trace with step-level confidence scores. We measure the self-correction success rate: the proportion of initially incorrect responses that the model successfully corrects after receiving feedback. Figure 4 shows that stepwise feedback leads to substantially higher self-correction success rates than answer-level feedback alone. The improvement is most pronounced for stronger reasoning models such as DeepSeek-R1 and Phi-4-Reasoning, which can better exploit localized error signals to revise their reasoning.

We also present a case study in Appendix H that illustrates how providing stepwise feedback enables an LLM to self-correct its errors. Since early mistakes can propagate and compound along the reasoning chain, we further report the performance of detecting the *first* error in Appendix I.

### 5.4. Ablation Study

To investigate the effectiveness of different design choices, we conduct ablation experiments on Phi4-Reasoning across three datasets. The mask predictor in GIBS receives input from both an edge encoder, which captures local structural information, and a graph encoder, which provides global context. As shown in Table 2, removing either component leads to a noticeable drop in AUROC, confirming that both local and global signals are essential for accurate SCA.

### 5.5. Generalization Analysis

**Generalization to Label-Free Settings** While our primary formulation utilizes final-answer correctness labels, practical deployment often lacks ground truth verification. We investigate whether our framework can generalize to such label-free scenarios by deriving consensus anchors from model-generated signals. We compare two unsupervised strategies: (1) *All Trajectories*, which builds consensus from all sampled traces, and (2) *Self-Consistency*, which replaces gold labels with majority-voted pseudo-correct trajectories. As shown in Table 3, using all trajectories suffers a clear performance drop due to noise from incorrect paths. The self-consistency variant, while still below the oracle *Correct-only* setting, achieves competitive results on Deepseek and Phi4. We observe that performance correlates with pseudo-label quality: on Phi4, self-consistent trajectories overlap approximately $80\%$ with gold labels, resulting in only a minor performance drop; for Llama3.1-8B, overlap drops to about $28\%$, with corresponding degradation. These

results indicate that our framework remains effective even without gold labels, as long as reasonably accurate reference trajectories can be obtained.

**Generalization to Out-of-distributions.** A practical CE method should remain effective beyond the domain it was trained on. To test this, we train GIBS with the trajectories generated by Phi4-Reasoning on MoreHopQA, a textual multi-hop QA dataset, as an in-distribution (ID) domain and evaluate it directly on the Math dataset without re-training as an out-of-distribution (OOD) domain. As shown in Figure 5, GIBS consistently achieves higher AUROC than NIBS and white-box baselines despite the domain shift. This advantage arises because NIBS relies on step similarity within the training domain, whereas GIBS learns structural representations under the IB principle, enabling it to capture abstract reasoning patterns that are less tied to domain-specific vocabulary. These results suggest that graph-based modeling provides strong robustness to domain variability.

## 6. Conclusion

This paper addresses stepwise confidence estimation in LLM reasoning, a key capability for diagnosing reasoning errors and enabling reliable model use. We introduced a framework for black-box LLMs based on the Information Bottleneck principle, with two complementary instantiations: NIBS, which assigns confidence via direct consensus alignment, and GIBS, which learns subgraph selection with consensus regularization. Across mathematical and textual reasoning benchmarks, both methods produced accurate, well-calibrated confidence scores that reliably localize erroneous steps. We further showed that stepwise confidence is actionable: integrating these signals into selective correction improved final-answer accuracy, and NIBS and GIBS demonstrated strong robustness and generalization.

## Acknowledgment

The work was partially supported by NSF award #2442477 and #2550203. Cheng is supported by the National Science Foundation (NSF) Grant #2312862, NSF-Simons SkAI Institute, NSF CAREER #2440542, NSF #2533996, National Institutes of Health (NIH) #R01AG091762, a Google Research Scholar Award, and Cisco gift grant. We thank Amazon Research Awards, Cisco Faculty Research Awards, and Toyota Faculty Research Awards. The authors acknowledge Google and OpenAI for providing us with API credits and Research Computing at Arizona State University for providing computing resources. The views and conclusions in this paper are those of the authors and should not be interpreted as representing any funding agencies.

## Impact Statement

The expected impact of this work is primarily within the research community, by providing insights that may help improve model inference. While large language models (LLMs) may have a wide range of societal implications depending on their downstream use, this paper is intended to serve as an auditing aid for human oversight on LLMs rather than a replacement, which does not directly enable applications that raise novel ethical concerns. There are many potential societal consequences of our work, none of which we feel must be specifically highlighted here.

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

# A. Notation Table

| Symbol | Description |
|---|---|
| $x$ | Input problem or question |
| $y_i = (T_i, A_i)$ | $i$-th sampled output: reasoning trajectory $T_i$ and final answer $A_i$ |
| $T_i = \{t_{i1}, t_{i2}, \ldots, t_{iL_i}\}$ | Reasoning trajectory with $L_i$ steps |
| $t_{ij} = (k_{ij}, a_{ij})$ | Step $j$ in trajectory $i$: description $k_{ij}$ and intermediate result $a_{ij}$ |
| $z_i \in \{0, 1\}$ | Correctness label of final answer $A_i$ |
| $c_{ij} \in [0, 1]$ | Confidence score assigned to step $t_{ij}$ |
| $\mathcal{S}$ | Sampled set of reasoning trajectories |
| $G_i = (V_i, E_i)$ | Graph representation of trajectory $T_i$ |
| $v_{ij}, e_{ij}$ | Node (intermediate result) and edge (reasoning operation) for step $t_{ij}$ |
| $G^{MC}$ | Consensus graph constructed from correct trajectories |
| $G^* \subseteq G_i$ | Subgraph selected by GIBS as compressed representation |
| $\mathbf{p}_\theta = \{p_{\theta,ij}\}$ | Soft selection mask predicted by the model |
| $\mathbf{m}_i = \{m_{ij}\}$ | Consensus mask indicating alignment with $G^{MC}$ |
| $X, Z, Y$ | Variables in IB formulation: input trajectory, compressed representation, and correctness |

*Table 4.* Notation summary used throughout the paper.

# B. Extended Problem Formulation and Discussion

In this section, we provide the formal definition of standard Answer-level Confidence Estimation (CE), which serves as the preliminary basis for our Stepwise framework.

Let an LLM be represented as a probabilistic model $\mathcal{M}$ that generates a response $y$ conditioned on input $x$. Answer-level CE aims to assign a reliability score to the final answer $A$. The definition of this score depends on the model's transparency:

- **White-box Setting (Open-weights):** For models where internal states are accessible, confidence can be defined directly from token probabilities, e.g., $C(x, A) = p(A|x; \mathcal{M})$.

- **Black-box Setting (Closed-source):** For proprietary models where token-level probabilities are unavailable, confidence must be inferred from the agreement or consistency among $N$ sampled outputs $\{A_1, A_2, \ldots, A_N\}$.

**Problem 2** (Answer-level CE). *Given an input $x$ and $N$ sampled answers $\{A_i\}_{i=1}^N$, the goal of answer-level CE is to learn a mapping*

$$f_{ans} : \{A_i\}_{i=1}^N \to \{c_i\}_{i=1}^N,$$

*where $c_i$ is the confidence score of answer $A_i$, estimated from observable signals such as self-consistency or semantic similarity among the sampled outputs.*

Our main paper extends this formulation to the granular level of reasoning steps (Problem 1), addressing the limitation that high answer-level confidence does not guarantee the correctness of individual reasoning steps.

**Discussion: Internal Certainty vs. Correctness-Oriented Reliability**   A key motivation for our Stepwise Confidence Attribution (SCA) framework is the conceptual distinction between *model certainty* and *objective reliability*. In standard UQ, confidence is often equated with the model's internal likelihood (or negative entropy). However, in complex reasoning tasks, this proxy is often ill-calibrated due to two possible phenomena: (1) *Confident Hallucinations:* A model may generate an incorrect reasoning step with high token probability if it follows a common misconception or linguistic pattern. (2) *The "Uncommon but Correct" Paradox:* A valid reasoning step might be creative or syntactically unique, resulting in low token probability. Relying solely on internal certainty would penalize such steps, treating valid variability as noise.

To address these limitations, we adopt a *correctness-oriented* view, defining confidence as "consistency with valid reasoning trajectories." One might ask: *Why not simply measure consistency across all sampled trajectories, regardless of correctness?* Our empirical analysis (and the logic of SCA) suggests that computing consensus over a mixed distribution of correct and incorrect traces introduces fundamental **ambiguity**. If the consensus metric (e.g., MCS size) is calculated against *all* generated paths, the algorithm cannot distinguish whether a step aligns with a dominant *correct* logic or a dominant *error*

mode (e.g., a common trap). As observed in our preliminary experiments, aligning with unfiltered trajectories causes the discrimination performance (AUROC) to drop significantly (often to near-random levels, $\sim 0.5$).

Therefore, establishing anchors from **correct-only** trajectories is not merely an implementation detail but a theoretical necessity. It ensures that the "confidence" score specifically measures attribution to success, allowing creative but valid steps to receive high scores as long as they appear in at least one verified reasoning path.

## C. Algorithm Details

### C.1. NIBS Algorithm.

The procedure of NIBS is shown in Algorithm 1. Given a set of sampled trajectories, we first split them into correct and wrong subsets using the final-answer labels. For a target trajectory, we score each step by comparing it to steps drawn from correct trajectories: a step receives higher confidence if it has strong semantic matches (under the chosen similarity metric) to steps in correct solutions. Within-trajectory similarities are aggregated (e.g., by max or mean) to form the step's score. This produces step-wise confidence without parameter learning and serves as a closed-form IB instantiation that compresses traces to consensus-supported steps.

---

**Algorithm 1** NIBS

    **Input:** Sampled trajectories $\mathcal{S} = \{(T_i, A_i, z_i)\}_{i=1}^N$
    **Output:** Stepwise confidence scores $\{c_{ij}\}$
    Partition $\mathcal{S}$ into $\mathcal{S}_{\text{correct}} = \{T_i \mid z_i = 1\}$ and $\mathcal{S}_{\text{wrong}}$
    **for** each trajectory $T_i = \{t_{i1}, \ldots, t_{iL_i}\}$ **do**
        **for** each step $t_{ij}$ in $T_i$ **do**
            Compute similarities $\{\text{sim}(t_{ij}, t')\}$ for all steps $t'$ in trajectories from $\mathcal{S}_{\text{correct}}$
            Aggregate within-trajectory similarities by $\text{Agg}(\cdot)$ (e.g., max or mean)
            Assign $c_{ij}$ by Eq. 2
        **end for**
    **end for**
    **Return** $\{c_{ij}\}$

---

### C.2. GIBS Algorithm.

The training procedure of NIBS is shown in Algorithm 2. We convert each trajectory into a directed reasoning graph whose steps are represented as edge–node pairs. Using only correct graphs, we construct a consensus graph (e.g., via maximum common subgraph) and align it to each training graph to obtain a consensus mask on steps. In detail, the value for the consensus mask $\mathbf{m}_i$ can be computed as:

$$m_{ij} = \frac{1}{|\mathcal{S}_{\text{correct}}|} \sum_k \mathbf{1}\big[v_{ij} \in \text{MCS}(G_i, G_k)\big],$$

where $\mathbf{1}[\cdot]$ is an indicator function. Thus, $m_{ij} = 1$ if the step consistently appears in all MCS matches with correct graphs, $m_{ij} = 0$ if it never appears, and intermediate values reflect partial support.

A trainable model outputs a soft selection mask over steps; the IB objective is approximated by the sum of a mask-entropy term (encouraging compression) and a cross-entropy term aligning the mask with the consensus anchors (encouraging relevance). We optimize the model parameters by gradient descent to produce calibrated step-selection probabilities.

The inference procedure is shown in Algorithm 3. Given a new trajectory's reasoning graph, the trained model predicts a soft mask over steps. Each step's inclusion probability is reported as its confidence score. Optionally, the soft-masked subgraph can be visualized or passed to downstream routines (e.g., selective correction) to prioritize low-confidence steps while preserving logically consistent parts of the reasoning.

---

**Algorithm 2** GIBS: Training

---

**Input:** Reasoning graphs $\{G_i = (V_i, E_i)\}_{i=1}^N$ with labels $\{z_i\}$, model $f_\theta$
**Output:** Trained parameters $\theta$
Partition $\{G_i\}$ into $\mathcal{G}_{\text{correct}} = \{G_i \mid z_i = 1\}$ and $\mathcal{G}_{\text{wrong}}$
Construct consensus graph $G^{MC}$ from $\mathcal{G}_{\text{correct}}$ (e.g., MCS aggregation)
**for** each epoch **do**
    **for** each $G_i$ **do**
        Represent each step as $t_{ij} = (v_{ij}, e_{ij})$
        Align $G_i$ with $G^{MC}$ to obtain consensus mask $\mathbf{m}_i = \{m_{ij}\}$
        Compute soft mask $\mathbf{p}_\theta = f_\theta(G_i)$ with $p_{\theta,ij} \in [0,1]$
        Form soft subgraph $G_i^* = G_i \odot \mathbf{p}_\theta$
        Compute loss by Equation 6
        Update $\theta$ by gradient descent
    **end for**
**end for**
**Return** $\theta$

---

**Algorithm 3** GIBS: Inference

---

**Input:** A new reasoning graph $G = (V, E)$, trained model $f_\theta$
**Output:** Step-wise confidence scores $\{c_j\}$
Represent each step as $t_j = (v_j, e_j)$
Compute soft mask $\mathbf{p}_\theta = f_\theta(G)$
**for** each step $t_j$ **do**
    Set confidence $c_j = p_{\theta,j}$
**end for**
Optionally form $G^* = G \odot \mathbf{p}_\theta$ for visualization or downstream use
**Return** $\{c_j\}$

---

## C.3. MCS Algorithm.

MCS algorithm first identifies semantically similar edge pairs between the two reasoning graphs using an NLI model, and selects a small set of high-scoring candidates as seeds. Starting from each seed, it expands the common subgraph iteratively via a BFS, adding only those edges and nodes whose textual entailment scores exceed the thresholds and whose mappings remain consistent. In this paper, we set both thresholds $\tau_v, \tau_e = 0.7$. Among all candidate expansions, the largest resulting subgraph is returned as the MCS along with the induced node and edge mappings. By integrating semantic similarity through entailment scores, the algorithm can align reasoning steps that are lexically different but semantically equivalent.

---

**Algorithm 4** FindMaximumCommonSubgraph

---

**Input:** Two reasoning graphs $G_1 = (V_1, E_1), G_2 = (V_2, E_2)$; reasoning texts $\mathcal{R}_1, \mathcal{R}_2$; thresholds $\tau_v, \tau_e$
**Output:** Common subgraph $G^{MC}$, node mapping $\pi_V$, edge mapping $\pi_E$
Generate candidate edge pairs $\mathcal{C} = \{(e_1, e_2) \mid \text{NLI(Entail)}(\mathcal{R}_1(e_1), \mathcal{R}_2(e_2)) \geq \tau_e\}$
Sort $\mathcal{C}$ by combined edge–node similarity, keep top-$K$ seeds
**for** each seed pair $(e_1, e_2) \in \mathcal{C}$ **do**
    Initialize subgraph $G^*$ with $(e_1, e_2)$, and mappings $\pi_V, \pi_E$
    Expand $G^*$ by BFS over neighbors of $e_1$:
        For each $e_1' = (x_1, y_1)$, find best match $e_2' = (x_2, y_2)$ with entailment $\geq \tau_e$
        If compatible with $\pi_V$, update $G^*$ and mappings
    Keep the largest $G^*$ found as the current best
**end for**
**return** $G^{MC}$ with similarity annotations, along with $\pi_V, \pi_E$

---

## C.4. Complexity Analysis of the Heuristic MCS Algorithm

We analyze the time and space complexity of the proposed MCS algorithm. Let $G_1 = (V_1, E_1)$ and $G_2 = (V_2, E_2)$ denote the two input reasoning graphs, and write $m_1 = |E_1|$, $m_2 = |E_2|$, and $N = m_1 m_2$ for the number of edge pairs. We denote by $\mathcal{T}_{\text{NLI}}$ the time required for a single forward pass of the underlying NLI model, i.e., the cost of computing one entailment-based similarity score between two text segments.

**Time complexity.** The proposed heuristic MCS algorithm proceeds in three phases.

**(1) Pairwise similarity estimation.** In the first phase, the algorithm iterates over the Cartesian product $E_1 \times E_2$ and, for each edge pair $(e_1, e_2)$, evaluates their semantic compatibility. This involves a constant number of calls to the NLI model to score the correspondence between the edges and their incident nodes. Thus the total cost of this phase is

$$O\big(N \cdot \mathcal{T}_{\text{NLI}}\big) \;=\; O\big(|E_1|\,|E_2| \cdot \mathcal{T}_{\text{NLI}}\big).$$

**(2) Candidate ranking and pruning.** Among all edge pairs that satisfy the similarity thresholds, the algorithm ranks them according to their combined edge–node similarity and retains only the top-$K$ pairs (with $K = 10$ in our experiments) as seed candidates. If we denote by $M \leq N$ the number of pairs that pass the thresholds, then sorting these candidates by score requires

$$O\big(M \log M\big) \;\subseteq\; O\big(N \log N\big) \;=\; O\big(|E_1|\,|E_2| \log(|E_1|\,|E_2|)\big)$$

time in the worst case. After sorting, truncating the list to the top-$K$ pairs is $O(1)$.

**(3) Heuristic subgraph expansion.** Starting from each of the at most $K$ seed pairs, the algorithm performs a BFS-style expansion on $G_1$, greedily adding neighboring edges and nodes as long as they admit compatible matches in $G_2$ within the pre-filtered candidate set. For a fixed seed, this expansion touches at most $O(|V_1| + |E_1|)$ graph elements of $G_1$, and since $K$ is a constant independent of the input sizes, the total cost of this phase is

$$O\big(K(|V_1| + |E_1|)\big) \;=\; O\big(|V_1| + |E_1|\big).$$

Combining these contributions, the overall running time of the heuristic MCS algorithm can be bounded as

$$\mathcal{T}_{\text{total}} \;=\; O\big(|E_1|\,|E_2| \cdot \mathcal{T}_{\text{NLI}} + |E_1|\,|E_2| \log(|E_1|\,|E_2|) + |V_1| + |E_1|\big).$$

In practice, the pairwise similarity phase is dominant, because it invokes computationally expensive NLI forward passes for $O(|E_1||E_2|)$ edge pairs. Treating the NLI architecture and the hyperparameter $K$ as fixed, the overall complexity with respect to the graph sizes is thus essentially quadratic in the number of edges:

$$\mathcal{T}_{\text{total}} = O\big(|E_1||E_2| \cdot \mathcal{T}_{\text{NLI}}\big).$$

**Space complexity.** The additional memory footprint is dominated by storing candidate edge pairs and the intermediate MCS structures. In the worst case, when many edge pairs pass the similarity thresholds, the algorithm keeps $M = O(|E_1||E_2|)$ candidates together with their scores. The data structures used for BFS-based expansion are linear in the size of $G_1$, i.e., $O(|V_1| + |E_1|)$, and are reused across different seeds. Therefore, the overall auxiliary space complexity is

$$\mathcal{O}_{\text{space}} = O\big(M + |V_1| + |E_1|\big) \subseteq O\big(|E_1||E_2|\big).$$

The parameters and internal buffers of the NLI model contribute only a constant term with respect to the graph sizes, and do not affect the asymptotic behavior.

# D. Experiment Details

## D.1. Datasets

We evaluate our methods on three verifiable reasoning datasets spanning both mathematical and textual reasoning tasks:

(1) **GSM8K** (Cobbe et al., 2021): A benchmark of grade-school math word problems, widely used for evaluating numerical reasoning and basic arithmetic logic.

(2) **Math** (Hendrycks et al., 2021): A more challenging dataset of competition-level mathematics problems. This dataset stresses the scalability of our methods to more complex mathematical reasoning.

(3) **MoreHopQA** (Schnitzler et al., 2024): A multi-hop question answering dataset, where solving a query requires integrating evidence across multiple passages. MoreHopQA focuses on textual compositional reasoning and tests whether our framework generalizes to non-mathematical domains.

## D.2. Baseline

We compare against four white-box CE baselines, each adapted to the step-wise reasoning setting:

(1) Normalized Sequence Likelihood (**SL(norm)**) (Lin et al., 2024; Cole et al., 2023). For each reasoning step $t_{ij}$, we concatenate the description $k_{ij}$ and the intermediate result $a_{ij}$ together. The step-wise confidence is then defined as the SL(norm) of all tokens in this step: $c_i^{SL} = \frac{1}{|T_i|} \sum_{t \in T_i} \log p_\theta(x_t|x_{<t})$.

(2) Token-level Entropy (**Entropy**) (Kuhn et al., 2023): Token Entropy computes the entropy of the predictive distribution at each token and then averages over tokens to obtain step-wise uncertainty, which is defined as $H_{\text{step}} = \frac{1}{|T|} \sum_{t=1}^{T} \sum_{k=1}^{V} p_{t,k} \log p_{t,k}$. Confidence is then taken as the $-H_{step}$.

(3) **P(true)** (Kadavath et al., 2022): Following prior work, we directly query the LLM itself to judge whether the current step is correct or not, and then use the probability assigned to the label *true* as the confidence score. The prompt we used is in Appendix D.5.

(4) **LeCo** (Yao et al., 2024): We also include LeCo, which introduces a logit-based confidence scoring method. It combines three components: the average token score, the step divergence score, and the inter-step transition score to compute an overall step confidence.

## D.3. Configurations

For all experiments, we evaluate three representative LLMs: LLaMA-3.1-8B-Instruct, Phi-4-Reasoning, and DeepSeek-R1-Distill-Qwen-32B, and use identical prompting templates (Appendix D.5) to ensure fair comparisons. Decoding is performed with temperature set to $1.0$, and for each input question, we sample $N = 20$ reasoning traces to balance the accuracy and diversity.

**Semantic Similarity.** To measure semantic similarity or equivalence, we experiment with two approaches commonly used

in UQ for LLMs: (1) cosine similarity between sentence embeddings from `bert-base-uncased` (Devlin et al., 2019), and (2) semantic entailment predictions from the `DeBERTa-large-MNLI` model (He et al., 2020). Our framework is flexible in this choice and can accommodate other similarity models.

**Reasoning Graph Extraction.** For GIBS, we extract reasoning graphs via structured generation: a LangFun-style template (Appendix D.5) asks the model to instantiate a Python `ReasoningGraph` class, and we extract nodes and edges with a simple rule-based parser. The edge- and node-level entailment thresholds $\tau_e, \tau_v$ used in the MCS procedure are both set to 0.7, following prior work (Da et al., 2025; Lin et al., 2023). A sensitivity analysis is also provided in Appendix E.

**GIBS Training.** We train GIBS on 2,000 reasoning graphs constructed from a mixture of correct and incorrect trajectories. Steps are encoded using embeddings from `bert-base-uncased`, and graph-level representations are obtained via a 2-layer GCN encoder with hidden dimension 128 and dropout 0.1. The model is optimized with Adam (learning rate $10^{-3}$) using early stopping based on validation loss.

**Evaluation.** Computing AUROC, AUCPR, and ACC@c% requires step-level correctness labels (not confidence ground truth), which are obtained from GPT-4o using the evaluation templates in Appendix D.5. All reported results are averaged over 10,000 reasoning traces per dataset. Experiments are implemented in PyTorch and run on a single NVIDIA A100 GPU.

## D.4. Evaluation Metrics

We evaluate how well the estimated confidence correlates with step-level correctness. We adopt three complementary metrics to evaluate the effectiveness of our approach: AUROC, AUCPR, and ECE.

(1) **AUROC** (Area Under the Receiver Operating Characteristic Curve) (Lin et al., 2023) measures the model's ability to discriminate between correct and wrong steps across all thresholds, reflecting overall ranking performance.

(2) **AUCPR** (Area Under the Precision–Recall Curve) (Davis & Goadrich, 2006) focuses on ranking under class imbalance, which is crucial since erroneous steps typically form a small fraction of all steps.

(3)**ACC@c%** (Geifman & El-Yaniv, 2017) is a standard metric in selective prediction, which reports the accuracy when retaining only the top-c% most confident predictions. A reliable CE method will get a higher ACC@c% by ranking correct predictions above incorrect ones.

(4) **ECE** (Expected Calibration Error) assesses the calibration of step-level confidence scores by comparing predicted confidence with empirical accuracy across bins, with lower values indicating better alignment.

## D.5. Prompt Template & Few Shot Examples

---

**Prompt Template for Structured LLM Responses**

**System Prompt:** Your role as an assistant involves thoroughly exploring questions through a systematic thinking process before providing the final, precise, and accurate solutions.

**Prompt:**
```
```
Please respond to the last INPUT_OBJECT with OUTPUT_OBJECT according to OUTPUT_TYPE.

INSTRUCTIONS:
- Do NOT define or repeat any class or function.
- ONLY produce an OUTPUT_OBJECT that instantiates the OUTPUT_TYPE.
- The output must be valid Python using the given type names.
- Do NOT generate code, explanation, or helper variables.
- Only output an object like ReasoningGraph(...).

INPUT_OBJECT:
1 + 1 =

OUTPUT_TYPE:
Answer

```python
```

---

```
class Answer:
final_answer: int
"'

OUTPUT_OBJECT:
"'python
Answer(
final_answer=2
)
"'

INPUT_OBJECT:
question

OUTPUT_TYPE:
ReasoningGraph

"'python
class ReasoningNode:
id: int
description: str
output: Union[int, float, str]
depends_on: list[int]

class ReasoningGraph:
nodes: list[ReasoningNode]
final_answer: Union[int, float, str]

OUTPUT_OBJECT:
"""
```

## Regeneration Template Final Answer Feedback

**Structured LLM Responses Prompt + Prompt:**
# NOTE: The previous final answer '{answer}' is incorrect.

Please regenerate the OUTPUT_OBJECT only.

Do not provide any reasoning, explanation, or extra text.

The OUTPUT_TYPE remains ReasoningGraph.

## Regeneration Template Stepwise Feedback

**Structured LLM Responses Prompt + Prompt:**
# FEEDBACK:

The previous final answer '{previous_answer}' is incorrect.

## Reasoning Process:

{reasoning_process}

## Identified Errors (steps with low confidence):

{error_description}

Do not follow the identified error steps. Please reanalyze and regenerate a correct final answer that is different from the previous incorrect answer which is {previous_answer}. Return only the ReasoningGraph object without any explanations.

*Table 5.* Parsing success rates of structured reasoning traces across datasets and models.

| Model | GSM8K | MoreHopQA | Math |
|---|---|---|---|
| Llama3.1-8B | 99.29% | 96.11% | 93.54% |
| DeepSeek-R1-Distill-Qwen-32B | 99.35% | 91.99% | 99.17% |
| Phi4-Reasoning | 99.69% | 98.23% | 97.92% |

| Method | GSM8K | | MoreHopQA | | Math | |
|---|---|---|---|---|---|---|
| | AUROC ↑ | Time ↓ | AUROC ↑ | Time ↓ | AUROC ↑ | Time ↓ |
| GIB-based | **0.7892** | **0.05h** | **0.6619** | **0.02h** | 0.6946 | **0.005h** |
| MCS-Only | 0.7743 | 19h | 0.6593 | 10h | **0.7254** | 5h |

*Table 6.* Comparison of GIB-based and MCS-Only methods w.r.t AUROC and inference time.

---

**GPT Evaluation Prompt**

**System Prompt:** You are evaluating a mathematical reasoning graph for correctness. Given the problem, correct answer, and a series of reasoning steps (edge-node pairs), determine if each step is mathematically and logically correct.

**Prompt:**
Math Problem: {question}
Correct Answer: {answer}
Reasoning Graph (Edge-Node Pairs):
{pairs_text}
Instructions:
1. Evaluate each edge-node pair in the context of solving this problem
2. Consider if the edge description accurately describes what is being calculated
3. Check if the node value is mathematically correct given the edge description
4. Verify that each step logically follows from the problem or previous steps
For each pair, respond with:
- 1 if the edge-node pair is correct
- 0 if the edge-node pair is incorrect
Format your response as a comma-separated list of digits (no spaces), one digit per pair, without any explanation.
Example for 5 pairs: 1,0,1,1,0
Your evaluation:

---

## D.6. Parsing success rate

We report the parsing success rate of structured reasoning traces in Table 5. Across datasets and models, the structured outputs can be parsed reliably. On GSM8K, all three models achieve over 99% parsing success. On MoreHopQA and Math, the rates also remain high, with Llama3.1-8B achieving over 93% across all datasets. These results indicate that modern instruction-tuned LLMs can reliably follow the structured output templates used in our framework.

## D.7. Inference Efficiency

A key advantage of GIBS is its computational efficiency at inference time. While our training objective encourages the selected subgraph to align with the consensus among correct solutions, computing MCS explicitly is expensive and requires access to high-quality correct solution sets.

GIBS addresses this by training with soft consensus regularization, allowing the model to internalize consensus patterns. At inference, GIBS directly predicts step-level confidence via a forward pass, bypassing MCS computation entirely. As shown in Table 6, GIBS achieves performance comparable to explicit MCS supervision while reducing inference time by over three orders of magnitude. This demonstrates that our approach effectively captures consensus structures without the computational overhead of exact MCS matching.

| Threshold | 0.5 | 0.6 | 0.7 | 0.8 | 0.9 |
|---|---|---|---|---|---|
| AUROC | 0.5821 | 0.6081 | 0.6619 | 0.6658 | 0.6112 |

Table 7. Sensitivity of AUROC to NLI thresholds on MoreHopQA with Phi-4.

| # Samples $N$ | 5 | 10 | 15 | 20 | 25 |
|---|---|---|---|---|---|
| AUROC | 0.5964 | 0.6024 | 0.6569 | 0.6619 | 0.6679 |

Table 8. Sensitivity of AUROC to the number of sampled trajectories $N$ on MoreHopQA with Phi-4.

## E. Sensitivity Study

We investigate the robustness of our framework with respect to two key hyperparameters: (i) the NLI-based similarity thresholds used in MCS construction, and (ii) the number of sampled trajectories $N$ used to form the consensus.

### E.1. Sensitivity of NLI thresholds.

In all main experiments, the entailment thresholds $\tau_e$ and $\tau_v$ are set to 0.7, following prior work on semantic alignment (Lin et al., 2023; Da et al., 2025). To assess sensitivity to this choice, we conduct a sensitivity analysis on MoreHopQA with Phi-4, jointly varying both thresholds in the range $[0.5, 0.9]$. As shown in Table 7, the AUROC remains stable when $\tau_e, \tau_v$ lie in $[0.7, 0.8]$, and we observe noticeable degradation only when the thresholds are set too strictly ($> 0.8$) or too loosely ($< 0.7$). This supports our default choice $\tau_e, \tau_v = 0.7$ as a robust operating point.

### E.2. Sensitivity of the number of sampled trajectories.

We also evaluate how performance depends on the number and diversity of sampled trajectories $N$ used for consensus construction. Table 8 reports AUROC on MoreHopQA with Phi-4 as we vary $N$ from 5 to 25. The results show that performance improves as $N$ increases and stabilizes once sufficient diversity is reached (approximately $N > 15$), indicating that the consensus becomes more reliable with richer sampling but remains robust beyond this point.

In principle, black-box confidence estimation methods approximate the underlying output distribution by sampling multiple trajectories; hence, performance naturally scales with $N$ before plateauing, a behavior consistently observed in prior work (Kuhn et al., 2023; Lin et al., 2023). Following these works and our empirical analysis, we set $N = 20$ in all main experiments as a good trade-off between computational efficiency and stability.

### E.3. Sensitivity of GNN backbones.

Our learned method, GIBS, uses a graph neural network as a backbone to encode consensus-aware features. In the main experiments, we adopt a standard 2-layer GCN. To assess the role of the backbone architecture, we also experimented with GraphSAGE, GAT, and GIN on MoreHopQA with Phi-4 under the same training protocol (Table 9). We find that all backbones yield comparable behavior in this setting, and a simple GCN already provides a strong and reliable choice for modeling the required structural patterns.

## F. Additional Evaluation on Stepwise Correctness

### F.1. Performance on PRM800K with Free-form Reasoning Traces

We evaluate our method on PRM800K (Lightman et al., 2023), a well-established stepwise reasoning benchmark containing pre-collected GPT-4 solutions with free-form chain-of-thought reasoning and high-quality human step-level annotations.

For each question, we sample $N = 20$ diverse traces and use the gold step labels. NIBS is applied in the same way as in the main experiments. For GIBS, each CoT is represented as a linear graph: each sentence-level step is treated as an edge, edges are connected sequentially, and node contents are left empty. We train on 2,000 traces and evaluate on 10,000 traces. Since PRM800K consists of pre-generated outputs from closed-source LLMs, token-level logits are unavailable, and white-box baselines cannot be applied. Since PRM800K consists of outputs from the closed-source model, token-level

| GNN Backbone | GCN | GraphSAGE | GAT | GIN |
|---|---|---|---|---|
| AUROC | 0.6619 | 0.6446 | 0.5806 | 0.5815 |

*Table 9.* Influence of different GNN backbones on AUROC (MoreHopQA with Phi-4).

| Method | AUROC ↑ | AUCPR ↑ | ACC@80% ↑ |
|---|---|---|---|
| White-box methods | N/A | N/A | N/A |
| Cos-Max | 0.6156 | 0.7840 | 0.7734 |
| Cos-Mean | 0.6821 | 0.8343 | 0.7860 |
| NLI-Max | 0.7666 | **0.9074** | 0.7899 |
| NLI-Mean | **0.8181** | 0.9019 | **0.8573** |
| GIBS | 0.6556 | 0.8203 | 0.7570 |

*Table 10.* Performance of NIBS and GIBS on the PRM800K dataset with pre-collected GPT-4 traces. White-box methods require token probabilities, which are not applicable in this setting.

logits are unavailable.

As shown in Table 10, both NIBS and GIBS achieve strong performance across metrics, demonstrating that our framework can adapt effectively to free-form CoT. NIBS variants perform best overall, while GIBS remains competitive but less dominant than in structured settings, reflecting the limited explicit structure available for GIBS to exploit.

### F.2. Performance on Larger Models

To evaluate whether SCA remains effective on larger models, we further conduct experiments with Qwen2.5-72B-Instruct on MoreHopQA. As shown in Table 11, GIBS consistently outperforms all baselines on both AUROC and AUCPR, achieving 0.7982 AUROC and 0.8385 AUCPR. These results indicate that the proposed graph-based stepwise attribution method remains effective when applied to stronger large-scale models.

### F.3. False Positive Analysis of Low-Confidence Steps

We further examine whether SCA incorrectly penalizes correct but uncommon reasoning steps. Specifically, we consider the bottom 20% lowest-confidence steps flagged by GIBS and compute the fraction of them that are actually correct, which serves as the false positive rate of low-confidence attribution.

As shown in Table 12, the false positive rates remain relatively low across datasets and models, ranging from 14.8% to 21.6%. This suggests that GIBS does not simply suppress uncommon reasoning patterns but more often assigns low confidence to genuinely problematic steps.

### F.4. Human Validation of Stepwise Attribution

To further validate the alignment between automatic attribution metrics and human judgment, we conduct a manual evaluation on MoreHopQA. For Phi4-Reasoning and DeepSeek-R1-Distill-Qwen-32B, we select 50 questions each and examine one incorrect trajectory per question. We then check whether the step with the lowest GIBS confidence matches the human-identified erroneous step.

For Phi4-Reasoning, whose AUROC is 0.66 in this setting, the lowest-confidence step matches the human-identified erroneous step in 60.0% of cases (30/50). For DeepSeek-R1-Distill-Qwen-32B, whose AUROC is 0.81, the match rate increases to 80.0% (40/50). This trend is consistent with the AUROC ranking, suggesting that our automatic metrics reasonably reflect the quality of step-level confidence attribution.

## G. Comparison with Adapted Black-Box Baselines

To our knowledge, there are no existing black-box methods that directly address stepwise confidence attribution for multi-step reasoning. However, several related approaches have been proposed for evaluating reasoning quality or detecting errors. We adapt two representative methods for comparison:

**ROSCOE** (Golovneva et al., 2022) restructures linear reasoning chains into Premise-Augmented Reasoning Chains by

*Table 11.* Stepwise correctness attribution performance on MoreHopQA with Qwen2.5-72B-Instruct.

| Method | AUROC | AUCPR |
|---|---|---|
| SL-norm | 0.4547 | 0.5218 |
| Entropy | 0.6114 | 0.6461 |
| LeCo | 0.5924 | 0.6973 |
| Cos-Max | 0.3136 | 0.5226 |
| Cos-Mean | 0.5722 | 0.6937 |
| NLI-Max | 0.6607 | 0.7637 |
| NLI-Mean | 0.6200 | 0.6835 |
| GIBS | **0.7982** | **0.8385** |

*Table 12.* False positive rates of low-confidence steps identified by GIBS. We consider the bottom 20% lowest-confidence steps and report the fraction of them that are actually correct.

| Model | GSM8K | MoreHopQA | Math |
|---|---|---|---|
| Llama3.1-8B | 15.9% | 16.9% | 21.6% |
| Phi4-Reasoning | 15.4% | 18.3% | 20.2% |
| DeepSeek-R1-Distill-Qwen-32B | 16.9% | 14.8% | 18.7% |

identifying premise links between steps, forming a directed acyclic graph. Originally designed to improve error identification by verifying each step under its premises, we adapt PARC by using its step-level verification scores as confidence scores.

**PARC** (Mukherjee et al., 2025) restructures linear reasoning chains into Premise-Augmented Reasoning Chains by identifying premise links between steps, forming a directed acyclic graph. Originally designed to improve error identification by verifying each step under its premises, we adapt PARC by using its step-level verification scores as confidence estimates.

As shown in Table 13, both NIBS and GIBS substantially outperform these adapted baselines across all models and metrics. ROSCOE variants, designed for holistic reasoning evaluation rather than step-level error localization. PARC performs better by leveraging premise dependencies, but still falls short of our consensus-based methods, likely because PARC focuses on local premise-step verification rather than global consensus alignment. These results suggest that methods designed for related but distinct purposes do not transfer well to stepwise confidence attribution, highlighting the need for dedicated approaches like ours.

## H. Case Study

Figure 6 provides a case study from the MoreHopQA dataset that qualitatively demonstrates the advantage of our method. Initially, the LLM produces an incorrect answer by focusing on the wrong entity (the magazine's founding year instead of the publisher's). Our method correctly identifies this flawed premise by assigning low confidence to the initial reasoning steps. While simple final-answer feedback is insufficient for the model to find this root error, our step-wise feedback explicitly flags the low-confidence steps. This targeted guidance prompts the model to reconsider its flawed premise, revise its reasoning trajectory, and arrive at the correct answer. This case illustrates that LLMs can effectively self-correct when their reasoning

| | Llama3.1-8b | | | Deepseek | | | Phi4 | | |
|---|---|---|---|---|---|---|---|---|---|
| | AUROC | AUCPR | ACC@80% | AUROC | AUCPR | ACC@80% | AUROC | AUCPR | ACC@80% |
| ROSCOE-ss | 0.477 | 0.517 | 0.503 | 0.408 | 0.584 | 0.528 | 0.421 | 0.603 | 0.542 |
| ROSCOE-sa | 0.524 | 0.565 | 0.541 | 0.492 | 0.600 | 0.612 | 0.488 | 0.594 | 0.613 |
| PARC | 0.589 | 0.608 | 0.511 | 0.637 | 0.686 | 0.659 | 0.606 | 0.676 | 0.650 |
| NIBS | 0.512 | 0.553 | 0.531 | 0.666 | 0.776 | 0.649 | 0.580 | 0.663 | 0.696 |
| GIBS | **0.647** | **0.669** | **0.560** | **0.808** | **0.835** | **0.705** | **0.661** | **0.686** | **0.705** |

*Table 13.* Comparison with adapted black-box baselines on MoreHopQA. ROSCOE-ss and ROSCOE-sa are adapted from reasoning chain evaluation metrics; PARC is adapted from premise-based error identification. Both NIBS and GIBS substantially outperform these baselines.

Context: "The Bellingham Review is an American literary magazine published by Western Washington University ... The magazine was established in 1977... Western Washington University was founded in 1886..." Question: "What is the closest palindrome number to when the publisher of Bellingham Review was founded?"

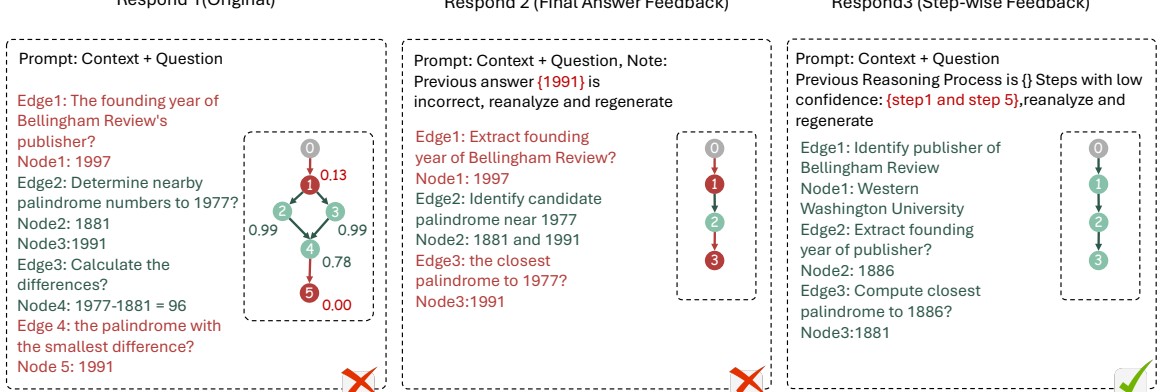

*Figure 6.* A case study on the MoreHopQA dataset comparing the effect of different feedback types. Providing targeted, step-wise feedback on low-confidence reasoning steps is effective at guiding the model to correct its root error, whereas providing simple final-answer feedback is not.

uncertainty is accurately localized.

## I. First-Error Step Detection

A natural concern is that step-wise confidence attribution might be dominated by *later* erroneous steps: once a trajectory has already gone off course, subsequent steps often become trivially inconsistent with the correct reasoning pattern. Thus, for debugging and test-time correction, identifying the *first* wrong step is most critical, since all downstream errors are typically propagated from this point.

Our framework inherently assigns confidence scores to *all* intermediate steps, so it also produces a score for the earliest erroneous step in each trace. To disentangle performance on the true failure point from that on subsequent steps, we conduct an additional evaluation that focuses exclusively on the first error. Concretely, for each trajectory that contains at least one incorrect step, we locate the first step whose correctness label is 0 and keep *only* this step when computing the metrics; all later steps in that trajectory are excluded from the evaluation set.

Table 14 summarizes the results on MoreHopQA for three LLMs. Across all models, GIBS achieves the best performance, indicating that it is particularly effective at ranking the first erroneous step above the correct ones.

| | Llama3.1-8b | Deepseek | Phi4 |
|---|---|---|---|
| | AUROC | AUROC | AUROC |
| P(true) | 0.5126 | 0.5152 | 0.5090 |
| SL(norm) | 0.4941 | 0.3009 | 0.4724 |
| Entropy | 0.5570 | 0.6004 | 0.6837 |
| LECO | 0.3839 | 0.3307 | 0.3884 |
| Cos-Max | 0.4275 | 0.4529 | 0.4772 |
| Cos-Mean | 0.4863 | 0.5313 | 0.5099 |
| NLI-Max | 0.5375 | 0.7110 | 0.6632 |
| NLI-Mean | 0.5285 | 0.6544 | 0.5824 |
| GIBS | **0.6164** | **0.7885** | **0.6841** |

*Table 14.* First-error step detection performance (AUROC) on MoreHopQA dataset.

