# OpenReview forum: "Diagnosing Multi-step Reasoning Failures in Black-box LLMs via Stepwise Confidence Attribution"
_ICML.cc/2026/Conference — ICML 2026 regular_

### Official Review · Reviewer_fFHw · 2026-03-05

**Soundness:** 3
**Presentation:** 3
**Significance:** 3
**Originality:** 3
**Overall Recommendation:** 5
**Confidence:** 4

**Summary:**

This paper introduces Stepwise Confidence Attribution (SCA), a framework for closed-source LLMs that assigns step-level confidence based only on generated reasoning traces. SCA applies the Information Bottleneck principle: steps aligning with consensus structures across correct solutions receive high confidence, while deviations are flagged as potentially erroneous. We propose two complementary methods: (1) NIBS, a nonparametric IB approach measuring consistency without graph structures, and (2) GIBS, a graphbased IB model that learns subgraphs through a differentiable mask to capture logical variability. Extensive experiments on mathematical reasoning and multi-hop question answering show that SCA reliably identifies low-confidence steps strongly correlated with reasoning errors.

**Compliance With Llm Reviewing Policy:**

Affirmed.

**Final Justification:**

My major concerns have been resolved.

**Key Questions For Authors:**

1. Why not use closed-source LLMs for expariments?
2. About the information bottleneck, I do not think there is a consensus structures across correct solutions. As shown in fig.1, soulutions B and C use different logics to solve the problem (One is based on bag, while the other is based on item).
3. Is the metrics reliable to reflect the quality of step-wise confidence attribution? Are they consistent with humans?

**Limitations:**

yes

**Strengths And Weaknesses:**

### Strengths
1. The research question is important.
2. The paper is well-writtern and easy to unsderstand.
3. The performance is good.

### Weaknesses
1. The experiments are limited to open-source LLMs, while the authors said the method is designed for closed-source LLMs.
2. The information bottleneck is not so motivated.

---

> ### Author Rebuttal · Authors · 2026-03-31
>
> **We are thankful for your time and constructive feedback. We were glad to hear your engagement with the technical details of our method.**
>
> We would like to provide a detailed clarification for each weakness and question.
>
> >W1 and Q1: Why are experiments limited to open-source LLMs?
>
> **Existing results on closed-source models:** In fact, we have evaluated on closed-source models: the PRM800K experiments (Appendix F) use pre-collected **GPT-4 traces** with no logits available, and both NIBS and GIBS achieve strong performance in this setting.
>
> **Why open-source models dominate the main experiments:** We clarify that "black-box" refers to the access mode (no token probabilities required), not open/closed-source status, and our methods are equally applicable to both. We chose open-source models for the main experiments specifically to enable comparison with other white-box stepwise baselines (e.g., SL(norm), Entropy), which require token-level logits unavailable from closed-source APIs.
>
> > W2: Motivation of the Information Bottleneck
>
> **Why IB:** Since step-level correctness labels are unavailable, we cannot directly supervise which step is wrong. Instead, our idea is to align a reasoning trace against all correct trajectories and infer step-level confidence from this alignment. The IB principle naturally formalizes this idea: it keeps the parts of a trace that are most informative for predicting correctness (relevance), while compressing away surface-level details that do not matter (compression).
>
> **How the two IB terms guide the design:** The compression term encourages the model to discard surface-level variation across reasoning traces and retain only a compact subset of steps that captures the common logical structure. The relevance term ensures that the retained steps remain informative about correctness. Under a natural simplifying assumption that steps consistently appearing across correct trajectories are the most informative about correctness, the IB objective admits a closed-form solution that is exactly NIBS: the compressed representation reduces to consensus steps, and confidence is computed directly from consensus alignment without training. GIBS relaxes this assumption by learning a parameterized mask: compression is achieved via a variational sparsification objective that pushes the mask toward selective, near-binary decisions, and relevance is approximated via a cross-entropy loss that aligns the predicted mask with consensus masks derived from correct reasoning graphs.
>
> > Q2: Solutions B and C in Fig. 1 use different logic; consensus across correct solutions may not exist.
>
> **Clarification:** The construction of consensus does not require all correct solutions to have the exact same logic structure. There are some diverse valid reasoning strategies as illustrated by Solutions B and C in Fig. 1. When evaluating a new solution, our method compares it against all correct trajectories in the sampled set (N=20), rather than requiring a match to any single one. A step receives higher confidence when it appears consistently across more correct paths, reflecting broader consensus support. Our multi-sampling strategy is designed precisely to capture diverse valid reasoning strategies.
>
> **Correct paths do share common substructures despite surface diversity:** Figure 3 empirically confirms this: correct trajectories concentrate near MCS proportion 0.8, showing that even when reasoning strategies differ in structure, correct solutions still share substantially more common substructures than incorrect ones.
>
> > Q3: Are the metrics reliable and consistent with human judgment?
>
> **Standard protocols:** These metrics are widely adopted evaluation protocols in confidence estimation literature [1,2,3].
>
> **Human validation:** To further validate alignment with human judgment, we conducted a manual evaluation on MoreHopQA. For Phi4 (AUROC=0.66) and DeepSeek (AUROC=0.81), we selected 50 questions each and examined one incorrect trajectory per question. The step with the lowest GIBS confidence matches the human-identified erroneous step in 60.0% (30/50) of Phi4 cases and 80.0% (40/50) of Deepseek cases. Notably, higher AUROC corresponds to greater human agreement, confirming that our automatic metrics reliably reflect step-level attribution quality.
>
> **Action 4.1:** We will include these human evaluation results in the revised paper.
>
> [1] Xiong, Miao, et al. "Can LLMs Express Their Uncertainty? An Empirical Evaluation of Confidence Elicitation in LLMs." The Twelfth ICLR.
>
> [2] Geng, Jiahui, et al. "A survey of confidence estimation and calibration in large language models." Proceedings of the 2024 NAACL (Volume 1: Long Papers). 2024.
>
> [3] Lin, Zhen, Shubhendu Trivedi, and Jimeng Sun. "Generating with Confidence: Uncertainty Quantification for Black-box Large Language Models." Transactions on Machine Learning Research.
>
> **`We'd be happy to further discuss should any concerns remain!`**

---

> > ### Author Rebuttal · Reviewer_fFHw · 2026-04-03
> >
> > Thanks for the rebuttal, my major concerns have been resolved. I will raise the score to reflect this.

---

> > > ### Author Response · Authors · 2026-04-04
> > >
> > > We thank the reviewer for acknowledging our contribution and for confirming that the major concerns have been resolved. Your feedback throughout this discussion has been very valuable!

---

### Official Review · Reviewer_faze · 2026-03-08

**Soundness:** 2
**Presentation:** 3
**Significance:** 3
**Originality:** 3
**Overall Recommendation:** 3
**Confidence:** 3

**Summary:**

This paper addresses the challenge of diagnosing errors within the multi-step reasoning traces of black-box Large Language Models (LLMs). The authors propose a framework called Stepwise Confidence Attribution (SCA), which aims to assign confidence scores to individual reasoning steps without requiring access to internal model weights or logits. The core intuition is based on the Information Bottleneck (IB) principle: correct reasoning traces tend to converge on specific logical invariants or "consensus structures," while incorrect traces diverge.

**Compliance With Llm Reviewing Policy:**

Affirmed.

**Final Justification:**

The authors have addressed most of my concerns, especially by clarifying the intended setting and adding results on larger models. I still have some concerns about the label-free setting for weaker models and robustness in harder cases. Thus, I will maintain my scores.

**Key Questions For Authors:**

1) How does your framework distinguish between a consensus of correctness and a consensus of error? Have you analyzed the failure cases where GIBS assigns high confidence to incorrect steps simply because the model was consistently wrong?
2) What is the parsing success rate for the models used (especially the smaller Llama-3-8B)?

**Limitations:**

The method requires sampling N=20 trajectories per query to build a reliable consensus. This increases the inference cost and latency by a factor of 20 compared to standard greedy decoding. While this is a common trait of consistency-based methods, the paper does not sufficiently discuss the trade-off between this high computational cost and the marginal gain in error diagnosis compared to cheaper, prompt-based verification methods.

**Strengths And Weaknesses:**

Strengths:
1) By parsing reasoning into nodes and edges and using Graph Neural Networks (GNNs) to capture dependencies, the method can better handle the non-linear nature of complex reasoning where step order might vary but logical dependencies remain constant.
2) Beyond just detection, the paper demonstrates the practical utility of the proposed metric. The experiments showing that stepwise feedback (derived from SCA) improves self-correction success rates by up to 13.5% over simple answer-level feedback provide tangible evidence that the confidence scores are meaningful and actionable.

Weaknesses:
1)  The fundamental assumption of the framework is the availability of a set of correct trajectories to form consensus anchors. In the main method, this requires ground-truth final labels to partition the sampled trajectories. While the authors attempt to address this in Section 5.5 using Self-Consistency as a proxy for correctness, the performance drop is significant for weaker models.
2) The experimental evaluation is limited to relatively smaller models (e.g., Llama-3-8B, Phi-4) and does not include larger LLMs.
3) NIBS ignores structural dependencies entirely, treating reasoning as bag-of-steps, yet reasoning is inherently sequential and compositional.

---

> ### Author Rebuttal · Authors · 2026-03-30
>
> **We are thankful for your time and constructive feedback, and we are glad you recognized the value of our framework.**
>
> We would like to provide a detailed clarification for each weakness, question, and limitation.
>
> >W1: Performance degrades for weaker models when using self-consistency as a label-free proxy.
>
> **Robustness:** We want to emphasize that the framework remains effective even in the worst-case scenario. For Llama3.1-8B under the label-free setting, GIBS achieves AUROC 0.5734, still outperforming the best white-box baseline on MoreHopQA (0.5510) and All Trajectories (0.5192). The framework degrades gracefully rather than failing.
>
> **Intended setting:** Our framework targets diagnostic scenarios where answer-level labels are cheap (exact match or automated verification), but step-level annotations are not. The core contribution is bridging this gap. The label-free analysis in Section 5.5 is an additional investigation beyond this intended setting.
>
> **Shared challenge:** This degradation is not unique to our method; it also affects other self-consistency-based approaches with smaller models. Our framework offers a practical advantage: practitioners can estimate the correctness of pseudo-labels to decide whether the consensus signal is sufficiently reliable for their use case.
>
> >W2: Limited model scale.
>
> **Clarification:** Our evaluation already includes **14B (Phi-4-Reasoning) and 32B (DeepSeek-R1-Distill-Qwen-32B)** scale models. Since our black-box framework relies solely on generated traces, scaling requires no architectural modifications.
>
> **New results:** To further address this concern, we conducted additional experiments on **Qwen2.5-72B-Instruct**. On MoreHopQA, GIBS achieves AUROC 0.7982, and NLI-Max achieves 0.6607, substantially outperforming other baselines SL(norm) (0.4547), Entropy (0.6114), and LeCo (0.5924).
>
> **Action 2.1:** We will include the 72B results in the revised paper.
>
> >W3: NIBS ignores structural dependencies and treats reasoning as a bag-of-steps.
>
> **Clarification:** NIBS is intentionally designed as a lightweight, training-free method. It is not a naive bag-of-steps approach: the aggregation strategies in Eq. 2 (max for local alignment, mean for global consistency) capture meaningful consensus signals, which is why NIBS achieves competitive results in Table 1.
>
> **Complementary design:** The compositional nature of reasoning is precisely why we proposed GIBS, which explicitly models structural dependencies via graph representations and consistently outperforms NIBS across all settings. NIBS offers a training-free, practical option; GIBS delivers full, structure-aware diagnostics when higher accuracy is needed.
>
> >Q1: How to distinguish consensus of correctness and error?
>
> **By Design:** Consensus anchors are constructed exclusively from trajectories with verified correct final answers, preventing erroneous consensus from wrong-answer paths. Table 3 validates this: using all trajectories causes substantial AUROC drops. For subtler errors within correct-answer trajectories, the IB compression term down-weights erroneous steps across multiple independent samples.
>
> **Failure Analysis:** Our manual analysis of 50 failed MoreHopQA questions shows that errors are particularly hard to distinguish when incorrect steps have high structural and semantic similarity to valid reasoning steps. These cases reflect genuine ambiguity in the reasoning content, rather than a consensus of error across trajectories.
>
> **Action 4.1:** We will include this failure analysis in the revised paper.
>
> >Q2:Parsing success rate
>
> **New Results:** On GSM8K, all three models achieve >99% success rate (Llama3.1-8B: 99.29%, DeepSeek-R1: 99.35%, Phi4: 99.69%). On MoreHopQA, rates remain high (Llama3.1-8B: 96.11%, DeepSeek-R1: 91.99%, Phi4: 98.23%). On Math, similarly (Llama3.1-8B: 93.54%, DeepSeek-R1: 99.17%, Phi4: 97.92%). Even Llama3.1-8B consistently achieves >93% across all datasets, confirming that parsing failures do not pose a practical bottleneck.
>
> **Action 5.1:** We will add these statistics to the appendix.
>
> >Limitation: Computational cost of multiply sampling
>
> **Correction: This reflects a misunderstanding of our inference procedure.** GIBS **does not** require multi-sampling at inference time. Once trained, it produces step-level confidence via a single forward pass (Algorithm 3). The N sampling is only needed during one-time training, and Table 5 shows that GIBS reduces inference time by over three orders of magnitude vs. explicit MCS computation. For NIBS, which does require sampling at inference, performance remains competitive with fewer samples (Table 7: N=15 achieves AUROC 0.6569 vs. 0.6619 at N=20). Regarding prompt-based methods: P(true) consistently underperforms across all settings (Table 1), demonstrating that the gain is substantial, not marginal.
>
> Due to space constraints, complete extended analyses can be provided in the revised paper or the next discussion period. Thanks again!

---

> > ### Author Rebuttal · Reviewer_faze · 2026-04-02
> >
> > The authors have made progress in addressing my concerns, yet the main weaknesses are not fully mitigated. I will therefore maintain my score.

---

> > > ### Author Response · Authors · 2026-04-04
> > >
> > > Thank you for engaging with our rebuttal and for noting that our responses have partially resolved your concerns. We appreciate your willingness to continue the discussion.
> > > **We noticed that you mentioned having follow-up questions, but did not list them explicitly. We would be very grateful if you could share your specific remaining concerns or follow-up questions, so that we can address them precisely rather than speculate.** In the meantime, we would like to proactively strengthen our response on the points most likely to remain open:
> > >
> > > >W1: Label Dependence and Weak-Model Degradation
> > >
> > > We understand this may be the central remaining concern, so we provide additional evidence and clarification.
> > >
> > > - **The label-free setting is harder than it appears, but our method still leads.** To put the Llama3.1-8B self-consistency result in a fuller context, we compare it against all baselines on MoreHopQA below. Even without any gold labels, GIBS with self-consistency outperforms every white-box method that has full access to token-level logits: This means that a practitioner with zero ground-truth labels and zero model internals still gets better step-level diagnostics from our method than any white-box approach. We believe this is a strong practical result.
> > >
> > > | Method | Access | AUROC |
> > > |---|---|---:|
> > > | P(true) | white-box | 0.5228 |
> > > | SL(norm) | white-box | 0.4005 |
> > > | Entropy | white-box | 0.5510 |
> > > | LECO | white-box | 0.3280 |
> > > | GIBS (self-consistency, no gold labels) | black-box | 0.5734 |
> > > | GIBS (cross-model consistency, no gold labels) | black-box | 0.6332 |
> > > | GIBS (gold labels) | black-box | 0.6471 |
> > >
> > > - **The "weak model" framing deserves nuance.** The performance gap between self-consistency and gold-label settings correlates directly with pseudo-label quality (~28% overlap for Llama3.1-8B vs. ~80% for Phi4). **This is not a flaw of our framework but a property of the input signal.** As models improve, which is the prevailing trend, pseudo-label quality increases and the gap closes naturally without any modification to our framework. Alternatively, for weaker models, pseudo-labels can be provided by a stronger model at minimal cost. We conducted a preliminary experiment using Phi4's majority-voted pseudo-labels to construct consensus anchors for Llama3.1-8B's trajectories on MoreHopQA. This cross-model setting improves AUROC from 0.5734 to 0.6332.
> > >
> > > - **In the intended diagnostic setting, labels are available by definition.** When a practitioner wants to diagnose why a model fails on a reasoning task, ground-truth answers are already part of the evaluation protocol. Our framework converts these freely available answer-level labels into step-level diagnostics that would otherwise require expensive human annotation. We see this as a significant practical contribution, not a limitation.
> > >
> > > >W3: NIBS ignores structural dependencies
> > >
> > > We appreciate this concern and would like to clarify our design rationale with additional evidence.
> > >
> > > - **NIBS is intentionally designed as a training-free, lightweight method that does not require an explicit graph structure** and treats reasoning as a chain of thought. As the reviewer noted, the order of reasoning steps is not fixed across solutions, and **the same logical operations can appear in different sequences.** In this situation, direct structural matching would contaminate effective semantic alignment, since topologically distant steps may be semantically equivalent and vice versa. The aggregation strategies are deliberate design choices that capture consensus signals while remaining robust to step ordering permutations. **The effectiveness of this design is empirically validated: on PRM800K (Appendix F)**, where reasoning traces are free-form chain-of-thought without explicit graph structure, NIBS substantially outperforms GIBS (NLI-Mean AUROC 0.8181 vs. GIBS 0.6556 in Table 9). This confirms that NIBS is not a missing structural modeling version of GIBS, but the more effective method when explicit graph structure is unavailable. This confirms that NIBS and GIBS are complementary components of our framework, allowing practitioners to choose based on their specific requirements and available resources.
> > >
> > > **`We look forward to your follow-up questions and are happy to provide any additional experiments or analyses that would help resolve your remaining concerns.`**

---

### Official Review · Reviewer_vLns · 2026-03-10

**Soundness:** 3
**Presentation:** 4
**Significance:** 3
**Originality:** 3
**Overall Recommendation:** 5
**Confidence:** 4

**Summary:**

This paper proposes a step-wise confidence attribution method based on the information bottleneck principle, quantifying the alignment between the step output with common structures across correct trajectories. It proposes a non-parametric method NIBS, and a learned method GIBS that additionally accounts for structural dependencies. Empirical results compared with white-box baselines on multiple reasoning datasets show that the proposed methods can identify errors. It further shows that step-wise attribution can enhance the model’s self-correction and accuracy. There are also ablations to label-free and out-of-distribution settings.

**Compliance With Llm Reviewing Policy:**

Affirmed.

**Final Justification:**

My concerns are resolved and I maintain my positive score.

**Key Questions For Authors:**

How are the ground-truth trajectories for the training data obtained?

**Limitations:**

Not explicitly discussed. See weaknesses.

**Strengths And Weaknesses:**

Strengths

1. The paper is well-written and well-presented. It is easy to read with clear documentation of the experiment setup and details.
2. The proposed step-wise error attribution methods in a black-box setting are novel, addressing a gap of existing methods which focus on white-box or final-answer, and have important practical implications
3. The empirical results are convincing in showing the effectiveness of the method. Ablations on generalization and demonstrations of the practicality of model self-correction are helpful.

Weaknesses

1. The method proposed is based on the hypothesis correlating confidence with alignment between steps and invariance across correct answers. While I agree that the empirical evidence supports the argument, I wonder how it accounts for rarely seen but correct steps (e.g., out-of-the-box thinking). What are the rates of false positives? Therefore, the method may potentially discourage model creativity in reasoning tasks.
2. The self-correction demonstration is not compared with other step-wise attribution baselines, which otherwise can further support the central hypothesis used in this paper to show its effectiveness over other frameworks.

---

> ### Author Rebuttal · Authors · 2026-03-30
>
> **We are thankful for your time and constructive feedback. We were glad to hear that you found the empirical evidence supportive of our central hypothesis.**
>
> We would like to provide a detailed clarification for each weakness and question.
>
> >W1: How does the method account for out-of-the-box thinking? What are the false positive rates?
>
> **Coverage through multi-sampling:** Our multi-sampling strategy (N=20) is designed precisely to capture diverse valid reasoning paths. As shown in Table 7, increasing N consistently improves performance with gains stabilizing around N=15–20, indicating that sampling progressively covers the space of valid strategies. Since our consensus mechanism aggregates across all correct-answer trajectories, a correct step appearing in even a small subset of sampled trajectories will still receive the consensus support. Only steps absent from all correct trajectories would receive low confidence. This situation serves as a diagnostic signal that this path is genuinely uncommon and warrants closer inspection, rather than discouraging creativity.
>
> **Low false-positive rates:** To evaluate whether GIBS incorrectly penalizes valid steps, we examine the steps flagged as least confident (the bottom 20% of confidence scores, consistent with our ACC@80% metric) and count how many are actually correct. Across all settings, this false positive rate ranges from 14.8% to 21.6%, confirming that GIBS rarely misidentifies correct steps as erroneous.
>
> | Benchmark | Llama3.1-8B | Phi4-reasoning | DeepSeek |
> |-----------|-------------|----------------|----------|
> | GSM8K     | 15.9%       | 15.4%          | 16.9%    |
> | MoreHopQA | 16.9%       | 18.3%          | 14.8%    |
> | Math      | 21.6%       | 20.2%          | 18.7%    |
>
> **Action 1.1:** We will include the false positive analysis in the revised paper.
>
> >W2: Self-correction is not compared with other step-wise attribution baselines.
>
> **Clarification:** The original comparison in Figure 4 was designed to demonstrate the value of step-level feedback over answer-level feedback, not to compare across attribution methods. To further address this concern, we provide self-correction results using different stepwise attribution methods on MoreHopQA with Phi4-Reasoning below.
>
> **New results:** With final-answer-only feedback, the correction success rate is 29.18%. Stepwise feedback from other baselines yields only marginal improvement. In contrast, GIBS achieves 42.69%, substantially outperforming all baselines.
>
> | Method   | Stepwise feedback |
> |----------|-------------------|
> | P(true)  | 29.89             |
> | SL(norm) | 29.95             |
> | Entropy  | 29.18             |
> | LECO     | 29.59             |
> | NLI-Max  | 30.12             |
> | GIBS     | 42.69             |
>
> **Action 2.1:** We will add the stepwise baseline comparison for self-correction to the revised paper.
>
> > Q1: How are the ground-truth trajectories for training data obtained?
>
> **No ground-truth trajectories needed:** Our framework does not require ground-truth trajectories or any human-annotated step-level labels. GIBS is trained using only final-answer correctness labels, which partition sampled trajectories into correct and incorrect sets. Step-level supervision (consensus mask) is derived automatically from the correct set via MCS aggregation (Appendix C). This is a key practical advantage of our approach.
>
> **`We'd be happy to further discuss should any concerns remain!`**

---

> > ### Author Rebuttal · Reviewer_vLns · 2026-04-02
> >
> > Thank you for the rebuttal. My concerns have been resolved and I maintain my positive rating.

---

> > > ### Author Response · Authors · 2026-04-04
> > >
> > > We thank the reviewer for the positive response and for confirming that the concerns are addressed. Your feedback throughout this discussion has been very valuable.

---

### Official Review · Reviewer_8ZoH · 2026-03-12

**Soundness:** 3
**Presentation:** 4
**Significance:** 4
**Originality:** 4
**Overall Recommendation:** 4
**Confidence:** 3

**Summary:**

This paper proposes Stepwise Confidence Attribution (SCA), a framework designed to diagnose multi-step reasoning failures in black-box LLMs without relying on internal model states or token probabilities. By leveraging the Information Bottleneck principle, the approach identifies logical invariants across sampled correct solutions to estimate step-level confidence. The authors introduce two complementary methods, a non-parametric approach (NIBS) and a learnable graph-based approach (GIBS), which effectively localize reasoning errors and improve self-correction.

**Compliance With Llm Reviewing Policy:**

Affirmed.

**Key Questions For Authors:**

/

**Limitations:**

yes

**Strengths And Weaknesses:**

**Strengths:**

1. Formulating stepwise confidence attribution based on the Information Bottleneck principle provides a theoretical foundation for isolating logical consensus.
2. The methodology uniquely targets black-box language models, making it practical for online LLM APIs where internal state access is restricted.
3. The dual introduction of a simple, training-free method (NIBS) and a more accurate, structure-aware model (GIBS) provides complementary solutions.

**Weaknesses:**

4. The self-correction experiment setup is limited, as it asks the model to regenerate its answer once without exploring the method's effectiveness in more complex, multi-turn interactions.
5. The process of forcing the language model to output structured reasoning graphs using strict code templates may be unstable and prone to parsing failures.

---

> ### Author Rebuttal · Authors · 2026-03-31
>
> **We are thankful for your time and constructive feedback. We were glad to hear your engagement with our experimental design and framework.**
>
> We would like to provide a detailed clarification for each weakness.
>
> > W1: The self-correction setup is limited to a single regeneration without exploring multi-turn interactions.
>
> **Clarification:** Our single-round setup was designed to isolate the effect of stepwise feedback versus answer-level feedback, which is the core contribution of our work. This controlled setting allows us to clearly demonstrate that fine-grained, step-level confidence signals lead to more targeted and effective corrections than answer-level feedback.
>
> **New multi-turn experiment:** Since GIBS can be directly applied during inference without any additional training and is easy to use in multi-turn interaction, we conducted an additional experiment where the model iterates up to 3 correction rounds, with GIBS re-applied at each round to update confidence scores and flag newly identified low-confidence steps. Results on MoreHopQA:
>
> | Model          | Round 1 | Round 2 | Round 3 |
> |----------------|---------|---------|---------|
> | Phi4-Reasoning | 42.7%   | 56.8%   | 58.04%  |
>
> The largest gain occurs in the first correction round. Subsequent rounds yield diminishing but still positive returns, confirming that GIBS remains effective across multiple turns.
>
> Action 1.1: We will include these multi-turn results in the appendix.
>
> >W2: Forcing the model to output structured reasoning graphs using strict code templates may be unstable and prone to parsing failures.
>
> **Empirical evidence of high reliability:** We provide parsing success rates across all models and datasets:
>
> |   | Llama3.1-8b | DeepSeek-R1-Distill-Qwen-32B | Phi4-reasoning |
> |------------|-------------:|-----------------------------:|---------------:|
> | GSM8K      | 99.29%       | 99.35%                       | 99.69%         |
> | MoreHopQA  | 96.11%       | 91.99%                       | 98.23%         |
> | Math       | 93.54%       | 99.17%                       | 97.92%         |
>
> These results confirm that modern instruction-tuned LLMs are highly capable of following structured output templates, and parsing failures do not pose a practical bottleneck for our framework.
>
> **Framework flexibility:** Importantly, our framework is not tied to any specific parsing strategy. As stated in Section 4.4, the LangFun-style prompting is one implementation choice. Our PRM800K experiments (Appendix F) demonstrate this directly: no structured template was used at all, yet both NIBS and GIBS still achieve strong performance. This highlights the generality of our approach beyond any particular output format.
>
> **` We'd be happy to further discuss should any concerns remain! `**

---

> > ### Author Rebuttal · Reviewer_8ZoH · 2026-04-02
> >
> > Thank you for the rebuttal and for conducting the new multi-turn experiment. I maintain my positive rating.

---

> > > ### Author Response · Authors · 2026-04-04
> > >
> > > We sincerely thank the reviewer for the positive response and valuable feedback throughout this discussion. If any follow-up questions remain, please let us know, and we are happy to discuss further during the remaining discussion period.

---

### Decision · Program_Chairs · 2026-04-30

**Decision:**

Accept (regular)

**Comment:**

The paper introduces Stepwise Confidence Attribution (SCA), a framework for diagnosing failures in multi-step reasoning traces under a black-box access setting. Rather than relying on token probabilities or model internals, the method uses generated reasoning traces and answer-level correctness labels to infer which intermediate steps are reliable. The central idea is that correct solutions may vary in surface form but tend to share consensus logical structures, while erroneous reasoning steps deviate from these structures. The paper instantiates this idea through two methods: NIBS, a non-parametric consensus-alignment approach, and GIBS, a learned graph-based method that represents reasoning traces as graphs and predicts step-level confidence via an information-bottleneck-inspired mask.

Reviewers found the problem important and the submission generally strong. They appreciated the focus on black-box step-level diagnostics, the complementary design of a simple training-free method and a more accurate graph-based method, the clear presentation, and the empirical evaluation on mathematical reasoning and multi-hop QA tasks. The results show that GIBS generally outperforms white-box confidence baselines in identifying erroneous steps, while NIBS remains a useful lightweight alternative. Reviewers also valued the self-correction experiments, which show that highlighting low-confidence steps improves correction success over answer-level feedback alone. Therefore I recommend acceptance.